# Corporate Needs You to Find the Difference: Revisiting Submodular and Supermodular Ratio Optimization Problems

**Elfarouk Harb**
University of Illinois at Urbana-Champaign
`eyfmharb@gmail.com`

**Yousef Yassin**
Carleton University
`yousef.yassin@carleton.ca`

**Chandra Chekuri**
University of Illinois at Urbana-Champaign
`chekuri@illinois.edu`

## Abstract

We consider the following question: given a submodular/supermodular set function $f : 2^V \to \mathbb{R}$, how should one minimize/maximize its average value $f(S)/|S|$ over non-empty subsets $S \subseteq V$? This problem generalizes several well-known objectives, including Densest Subgraph (DSG), Densest Supermodular Set (DSS), and Submodular Function Minimization (SFM). Motivated by recent applications [42, 34], we formalize two new broad problems: the Unrestricted Sparsest Submodular Set (USSS) and Unrestricted Densest Supermodular Set (UDSS), both of which allow negative and non-monotone functions.

Using classical results, we show that DSS, SFM, USSS, UDSS, and Minimum Norm Point (MNP) are all equivalent under strongly polynomial-time reductions. This equivalence enables algorithmic cross-over: methods designed for one problem can be repurposed to solve others efficiently. In particular, we use the perspective of the minimum norm point in the base polyhedron of a sub/supermodular function, which, via Fujishige's results, yields the dense decomposition as a byproduct. Through this perspective, we show that a recent converging heuristic for DSS, SUPERGREEDY++ [17, 32], and Wolfe's minimum norm point algorithm are both universal solvers for all of these problems.

On the theoretical front, we explain the observation made in recent work [42, 34] that SUPERGREEDY++ appears to work well even in settings beyond DSS. Surprisingly, we also show that this simple algorithm can be used for Submodular Function Minimization, including acting as a practical minimum $s$-$t$ cut algorithm.

On the empirical front, we explore the utility of several algorithms for recent problems. We conduct over 400 experiments across seven problem types and large-scale synthetic and real-world datasets (up to $\approx 100$ million edges). Our results reveal that methods historically considered inefficient, such as convex-programming methods, flow-based solvers, and Fujishige-Wolfe's algorithm, outperform state-of-the-art task-specific baselines by orders of magnitude on concrete problems like HNSN [42]. These findings challenge prevailing assumptions and demonstrate that with the proper framing, general optimization algorithms can be both scalable and state-of-the-art for supermodular and submodular ratio problems.

39th Conference on Neural Information Processing Systems (NeurIPS 2025).

# 1 Introduction and Background

Submodular and supermodular functions play a fundamental role in combinatorial optimization and, thanks to their generality, capture a wide variety of highly relevant problems. For a finite ground set $V$, the real-valued set function $f : 2^V \to \mathbb{R}$ is submodular iff $f(A) + f(B) \geq f(A \cap B) + f(A \cup B)$ for all $A, B \subseteq V$. A set function $f$ is supermodular iff $-f$ is submodular; $f$ is normalized if $f(\emptyset) = 0$ and monotone if $f(A) \leq f(B)$ whenever $A \subseteq B$. When working with these functions, we typically assume they are available through a value oracle that returns $f(S)$ given a set $S \subseteq V$. We make this assumption throughout the paper. A problem of central interest is the following:

**Problem 1.** *(Submodular Function Minimization, SFM). Let $f : 2^V \to \mathbb{R}$ be a submodular function (not necessarily monotone) given via a value oracle. Compute $\min_{\emptyset \neq S \subseteq V} f(S)$.*

A classical result in combinatorial optimization is that SFM can be solved in strongly polynomial-time [30]. There have been many theoretical developments since then [37, 48, 51, 35, 12, 39, 3].

In this paper, we are interested in *ratio* problems involving submodular and supermodular functions, which have been very interesting in various applications. We start with a concrete and canonical problem of this form.

**Problem 2.** *(Densest Subgraph, DSG). Given an undirected graph $G = (V, E)$ find a subset $S \subseteq V$ that maximizes the density $|E(S)|/|S|$ where $E(S) = \{(u, v) \in E : u, v \in S\}$.*

DSG is a classical problem with wide-ranging applications in data mining, network analysis, and machine learning. Dense subgraphs often reveal crucial structural properties of networks and thus DSG has been a very active area of recent research; see, for example, [14, 44, 9, 19, 55, 54, 1, 56, 23, 45, 50, 6, 40, 2, 53, 43, 41, 47, 13, 7]. A key feature of DSG is its polynomial-time solvability. There are several algorithms to solve it exactly: (i) via network flow [29, 49], (ii) via reduction to SFM (folklore), and (iii) via an LP relaxation [15]. Despite these exact algorithms, there has been considerable interest in fast approximation algorithms and heuristics to scale to the large networks that arise in practice. Hence, the Greedy peeling algorithm that yields a $1/2$-approximation [15] has been popular in practice. Furthermore, there are several theoretical algorithms that obtain a $(1 - \varepsilon)$-approximation in near-linear time for any fixed $\varepsilon$ via different techniques [4, 10, 17]. An important recent algorithm, Greedy++, is an iterative version of Greedy that was proposed in [9]. It is simple, combinatorial, and has strong empirical performance. Moreover, it was conjectured to converge to a $(1 - \varepsilon)$-approximation in $O(1/\varepsilon^2)$-iterations, each of which takes (near) linear time like Greedy. [17] proved that Greedy++ converges to a $(1 - \varepsilon)$-approximate solution in $O(\Delta(G) \log |V|/\varepsilon^2)$ iterations where $\Delta(G)$ is the maximum degree. Crucial to their proof was a perspective based on supermodularity. In particular, they considered the following general ratio problem.

**Problem 3.** *(Densest Supermodular Set, DSS). Let $f : 2^V \to \mathbb{R}_{\geq 0}$ be a normalized, monotone supermodular function. Compute $\max_{\emptyset \neq S \subseteq V} f(S)/|S|$.*

DSG is a special case of DSS; for all graphs $G = (V, E)$, the function $f : 2^V \to \mathbb{R}$ where $f(S) = |E(S)|$ is monotone supermodular. DSS can be solved exactly via SFM. [17] defined SuperGreedy and SuperGreedy++ for DSS and showed that SuperGreedy++ converges to a $(1 - \varepsilon)$-approximate optimum solution in $O(\alpha_f \log |V|/\varepsilon^2)$ iterations where $\alpha_f = \max_v(f(V) - f(V - v))$. This is a useful result since several non-trivial problems in dense subgraph discovery can be modeled as a special case of DSS including hypergraph density problems, $p$-mean density [57] and others — for details, we refer the reader to the recent survey [41]. Several other converging iterative algorithms for DSG have recently been developed via convex optimization methods such as Frank-Wolfe [19], FISTA [31], and accelerated coordinate descent [47]. In another direction, flow-based exact algorithms have been revisited [34, 33]. Many of these algorithms are based on structural properties of DSG inherited from supermodularity, which we describe in more detail following the discussion of the motivation for this work.

**Motivation for this work:** Our initial motivation originates from two recent ratio problems. [42] studied the following problem.

**Problem 4.** *(Heavy Nodes in a Small Neighborhood, HNSN). Given a bipartite graph $G(L, R, E)$ and a weight function $w : R \to \mathbb{R}_{\geq 0}$, find a set $S \subseteq R$ of nodes such that $\sum_{v \in R} w(v)/|N(S)|$ is maximized where $N(S)$ is the set of neighbors of $S$. Equivalently the goal is to minimize $|N(S)|/(\sum_{v \in R} w(v))$.*

In the preceding problem, the function $f : 2^V \to \mathbb{R}$ defined as $f(S) = |N(S)|$ is a monotone submodular function; it is the coverage function of the set system induced by the bipartite graph $G$ (which can also be viewed as a hypergraph). This leads us to consider the following ratio problem.

**Problem 5.** *(Unrestricted Sparsest Submodular Set,* USSS*). Given a normalized submodular function $f : 2^V \to \mathbb{R}$, find $\min_{\emptyset \neq S \subseteq V} f(S)/|S|$.*

A second motivating problem is the following, which is initially studied in [18], and later revisited in [34].

**Problem 6.** *(Anchored Densest Subgraph,* ADS*). Given a graph $G = (V, E)$ and a set $R \subseteq V$, find a vertex set $S \subseteq V$ maximizing $\left(2|E(S)| - \sum_{v \in S \cap \overline{R}} deg_G(v)\right)/|S|$, where $deg_G(v)$ is the degree of $v$ in $G$.*

The numerator in the preceding problem is supermodular but is no longer non-negative or monotone. This motivates us to further define an unrestricted version of DSS.

**Problem 7.** *(Unrestricted Densest Supermodular Set,* UDSS*). Given a normalized supermodular function $f : 2^V \to \mathbb{R}$, find $\max_{\emptyset \neq S \subseteq V} f(S)/|S|$.*

[42] showed that the HNSN problem can be reformulated as a special case of DSS, allowing iterative peeling algorithms (SUPERGREEDY++) to converge to the optimal solution. In [34], this perspective is extended by applying SUPERGREEDY++ to the unrestricted version of DSS (UDSS), noting that non-negativity and monotonicity can be enforced if one first shifts the function by a large modular term, i.e., adding $C|S|$ for some sufficiently large constant $C$. This ensures non-negativity and monotonicity while retaining the exact optimal solution (since the density of all sets simply shifts by a constant $C$). However, the shift renders the relative approximation guarantee in [17] inapplicable in a direct way; indeed, [34] state the following: "we hypothesize that iterative peeling remains an effective practical heuristic".

More generally, while USSS and UDSS are equivalent in the exact optimization sense (via negation), approximation guarantees do not easily carry over, especially when relying on relative error measures. This highlights a key open question: can we formally prove that SUPERGREEDY++ converges for all the above problem classes under appropriate approximation guarantees? Addressing this question, as we will in this paper, would provide a unified theoretical foundation for its empirical success observed across diverse submodular and supermodular ratio problems.

**Min-norm point and the Fujishige-Wolfe algorithm.** Another motivation for this work comes from some essential properties of the min-norm point problem relevant to submodular optimization.

**Problem 8.** *(Minimum Norm Point,* MNP*). Let $f : 2^V \to \mathbb{R}$ be a normalized submodular or supermodular function, and let $B(f)$ be the corresponding base polytope. Find $\arg\min_{x \in B(f)} \|x\|_2^2$.*

It is known that SFM can be reduced to MNP [26, 52]. Further, [25, 26] showed that the optimum value yields a lexicographically optimal and unique base, which reveals the entire dense decomposition of $f$. Recent work in the context of DSG has exploited MNP via convex optimization techniques [19, 31, 47] to obtain fast iterative algorithms for DSG and the dense decomposition version of DSG. Wolfe defined the min-norm point problem in the more general context of convex optimization and developed an iterative algorithm which is well-known [59] — we note that his algorithm is tailored to the quadratic norm objective and is quite different from the more general convex optimization algorithms. Fujishige tailored the algorithm to submodular polytopes, yielding one of the fastest practical SFM solvers [28, 11]. Nagano *et al.* [46] also studied the densest $k$ subgraph problem in the context of the minimum norm point in the base polyhedron. Despite its well-known empirical performance for SFM, the Fujishige-Wolfe algorithm has not been empirically evaluated for any recent ratio problems.

**Revisiting flow-based exact algorithms.** Flow-based approaches to DSG were deemed impractical for large graphs because they required an expensive binary search procedure [1]. Recent independent work by Huang *et al.* [34] and Hochbaum [33] introduced an iterative *density-improvement*

---

[1]Binary search is efficient in general, and much faster than simply enumerating all possible density thresholds. Our intention here is to indicate that earlier flow-based algorithms for densest subgraph problems used binary search over the density threshold $\lambda$, requiring a full max-flow computation at each step of the search. The need for high-precision convergence and repeated max-flow computations can makes this approach impractical at scale.

framework that overcomes this limitation. Letting $f(S) = |E(S)|$, and starting with $S_0 = V$ and $\lambda_0 = f(S_0)/|S_0|$, the method repeatedly computes $S_{k+1} = \arg\max_{S \subseteq S_k} \{f(S) - \lambda_k|S|\}$ via max-flow, updating $\lambda_{k+1} = f(S_{k+1})/|S_{k+1}|$ until convergence ($S_{k+1} = \bar{S}_k$). This approach, rooted in Dinkelbach's classical method from 1967 [20, 34], guarantees optimality. Although this process may appear worse than binary search in the worst case, potentially requiring up to $|V| + 1$ flow computations, empirical evidence shows that the number of iterations is typically minimal and often far fewer than those required by binary search. These insights have renewed interest in flow-based algorithms for solving DSG. One can reduce HNSN to a max-flow problem. This raises the question of the performance of flow-based algorithms for HNSN.

**Summary of Motivation**. We are motivated by the observation that several closely related submodular and supermodular ratio problems are treated disparately across the literature, often with distinct algorithms and analyses, despite underlying connections. We are interested in whether a unified perspective can bridge these gaps. For instance, while the simple combinatorial algorithm SUPERGREEDY++ has provable guarantees for DSS, its empirical success in broader settings lacks a corresponding unified theoretical analysis.

Similarly, the Fujishige-Wolfe algorithm, despite its established effectiveness for submodular function minimization, has surprisingly not been evaluated for ratio problems in USSS and UDSS. Moreover, with recent progress in flow-based exact algorithms for densest subgraph problems, it is natural to ask how these methods perform on related submodular ratio problems like HNSN. In essence, our goal is to investigate whether algorithmic successes in one domain can be transferred to others, leveraging tools that have been overlooked to improve empirical performance and deepen theoretical understanding across these problem classes.

**Our Contributions.** We formalize connections between five problems via reductions: Minimum Norm Point (MNP), Submodular Function Minimization (SFM), Densest Supermodular Set (DSS), Unrestricted Sparsest Submodular Set (USSS), and Unrestricted Densest Supermodular Set (UDSS) by proving that they are **equivalent**. All the reductions run in either $O(n)$ or $O(n \log n)$ (where $n$ is the dimension of the underlying base set) calls and are highly efficient.

While these reductions are rooted in classical concepts, several of these problems—such as USSS and UDSS—have not been formally defined in the literature. For instance, although DSS can be exactly solved via SFM, its formal definition in [17] was instrumental in advancing specialized algorithms like SuperGreedy and SuperGreedy++. By establishing these reductions, we enable **algorithmic cross-over**, where methods developed for one problem can be effectively applied to others. This is demonstrated in our experiments: Fijishige-Wolfe's Minimum Norm Point algorithm (FW-MNP), originally for MNP, achieves up to $595\times$ speedups over prior state-of-the-art baselines on HNSN (a special case of USSS). Similarly, SUPERGREEDY++, designed for DSS, delivers scalable and competitive performance on the minimum $s$-$t$ cut problem (a special case of SFM). Interestingly, it is surprising that SUPERGREEDY++ serves as an effective algorithm for minimum $s$-$t$ cut, despite being extremely simple, requiring no special data structures, and working in the dual (cut) space. To the best of our knowledge, all existing algorithms for minimum $s$-$t$ cut in the literature operate in the primal space via max-flow formulations.

Additionally, we demonstrate that an approximate solution to the Minimum Norm Point (MNP) problem also provides approximate solutions to all the other problems. We use the notion of *additive* approximation rather than the relative approximation notion used in the DSG and DSS context. This is necessary since we are working with unrestricted functions that may be negative. We also prove that SUPERGREEDY++, the Frank-Wolfe algorithm, and the Fujishige-Wolfe FW-MNP Algorithm are, in fact, all (under the hood) solving the Minimum Norm Point problem, which explains why these methods have seen so much success across seemingly unrelated problems.

It is worth highlighting an interesting observation about SUPERGREEDY++. Although the algorithm was originally developed for the DSS problem— a task that itself reduces to SFM —our results show that SUPERGREEDY++ can also be *directly* interpreted as an algorithm for solving SFM. This "full-circle" insight, where an approximation algorithm for a derived problem effectively addresses a more general problem, is surprising and conceptually satisfying.

We summarize our contributions more concretely:

1. **Problem Equivalence.** We prove that the following problems can be reduced to one another in strongly polynomial time using efficient reductions when solved exactly: Submodular Function

Minimization (SFM), Densest Supermodular Set (DSS), Minimum Norm Point (MNP), Unrestricted Sparsest Submodular Set (USSS), and Unrestricted Densest Supermodular Set (UDSS).

2. **Approximation Transfer.** We prove that approximate solutions to MNP can be transformed into approximate solutions for all the above problems via approximation-preserving reductions. These approximation-preserving reductions are different from the above exact reductions.

3. **SuperGreedy++ and Fujishige-Wolfe FW-MNP Algorithm as Universal Solvers.** Despite being designed for specific settings, we prove both SUPERGREEDY++ and Fujishige-Wolfe's FW-MNP algorithm are universal solvers for this broader class of problems.

4. **Efficient New Algorithms for HNSN and Minimum $s$-$t$ Cut.** As a direct consequence, our results imply that the Unrestricted Sparsest Submodular Set, a primary motivation of our work, can be efficiently approximated and solved. This concretely includes, for instance, new faster algorithms for the HNSN problem that are orders of magnitude faster than all 6 state-of-the-art baselines compared in [42]. Additionally, we show that SUPERGREEDY++ is, in fact, an efficient submodular function minimization algorithm and can thus be used to solve the classical minimum $s$-$t$ cut problem, often converging to the minimum cut within a few iterations.

5. **Empirical Validation at Scale.** We conduct an extensive experimental study involving over 400 trials run across seven distinct problems, encompassing both synthetic and real datasets containing graphs with up to $\approx 100$ million edges or elements for set functions. Our experiments systematically compare all major algorithmic paradigms, including flow-based methods, LP solvers, convex optimization techniques (e.g., Fujishige-Wolfe's FW-MNP algorithm, Frank-Wolfe), and combinatorial baselines (e.g., SUPERGREEDY++), for each problem class: DSG, DSS, USSS, UDSS, SFM, and MNP. Notably, many algorithms that had never been previously explored in certain problem settings (e.g., Fujishige-Wolfe's FW-MNP algorithm on HNSN or flow-based methods on USSS instances) outperform problem-specific state-of-the-art algorithms from prior work by orders of magnitude.

Our results demonstrate that methods often written off as inefficient, such as Wolfe's algorithm and max-flow solvers, are not only competitive but frequently faster and more accurate than tailored heuristics *when applied correctly with the right lens*.

**Remark on Scope and Related Work**. Throughout this paper, we discuss several optimization problems such as SFM, HNSN, DSS, DSG, and MNP, each of which has a rich history and extensive literature. Due to space constraints, we cannot provide an exhaustive survey or explore all technical nuances and historical developments of each problem. Instead, we have focused on presenting the key connections and ideas necessary for the unified perspective proposed in this work. An expanded version of this paper will include a more comprehensive discussion, including detailed related work and contextual background. Due to space constraints, we include several proofs and experimental results in the Appendix. Finally, we include a table of acronyms in Table 2.

## 2 Preliminaries and Main Results

Recall that we already defined submodularity, supermodularity, and monotonicity in the introduction. Throughout this paper, we assume that all set functions are normalized without loss of generality.

**Definition 9** (Base Polymatroid). For a submodular function $f : 2^V \rightarrow \mathbb{R}$, the *base polymatroid* is defined as $B(f) = \{x \in \mathbb{R}^{|V|} : x(S) \leq f(S) \quad \forall S \subseteq V, \quad x(V) = f(V)\}$, where $x(S) = \sum_{u \in S} x_u$.

**Definition 10** (Base Contrapolymatroid). For a supermodular function $f : 2^V \rightarrow \mathbb{R}$, the *base contrapolymatroid* is defined as $B(f) = \{x \in \mathbb{R}^{|V|} : x(S) \geq f(S) \quad \forall S \subseteq V, \quad x(V) = f(V)\}$.

For functions that are either submodular or supermodular, we refer collectively to the base polymatroid and the base contrapolymatroid as the *base polytope*.

**Importance of the Minimum Norm Point in the Base Polytope.** A key approach in submodular optimization connects continuous minimization over the base polytope with discrete problems. Specifically, the MINIMUM-NORM-POINT (MNP) problem minimizes $\|x\|_2^2$ over $x \in B(f)$, yielding a unique minimizer $x^*$. Thresholding $x^*$ often recovers combinatorial minimizers for penalized objectives. While the following connection is classical in submodular function minimization literature, its utility for submodular or supermodular ratio problems has been underexplored. Its significance became clear to us only after recognizing how it simplifies several prior results.

**Lemma 11.** *Let $f$ be a normalized submodular set function and let $x^*$ be the optimal solution of $\min_{x \in B(f)} \|x\|_2^2$. For any $\lambda \in \mathbb{R}$, define the set $S_\lambda = \{v \in V : x_v^* \leq \lambda\}$. Then, $S_\lambda$ is a minimizer of the function $f(S) - \lambda|S|$.*

**Discussion.** Lemma 11 has strong implications, helping establish the equivalence of several problems through efficient, strongly polynomial reductions. Specifically, it shows (together with other ideas) that all the following problems are equivalent, with very efficient strongly-polynomial time reductions. *Proof in Appendix A.2.*

**Theorem 12.** *The following problems are all equivalent **when solved exactly**, with efficient (strongly polynomial) near-linear time reductions between them: (1)* MINIMUM NORM POINT (MNP) *(2)* SUBMODULAR FUNCTION MINIMIZATION (SFM) *(3)* DENSEST SUPERMODULAR SET (DSS) *(4)* UNRESTRICTED SPARSEST SUBMODULAR SET (USSS) *(5)* UNRESTRICTED DENSEST SUPERMODULAR SET (UDSS).

**Approximation equivalence.** Although Theorem 12 shows that the aforementioned problems are efficiently reducible to one another when solved exactly, this does not imply that the reductions are approximation-preserving. However, as the following Theorem demonstrates, it suffices to focus on approximating the MINIMUM NORM POINT problem (MNP).

**Theorem 13.** *Let $f : 2^V \to \mathbb{R}$ be a normalized submodular or supermodular function, and suppose we can compute $\hat{x} \in B(f)$ satisfying*

$$\|\hat{x}\|_2^2 \leq \langle q, \hat{x} \rangle + \varepsilon^2 \quad \text{for all } q \in B(f),$$

*i.e., an upper bound on the duality gap. Let $n = |V|$. Then, $\hat{x}$ can be used to efficiently obtain approximate solutions:*

1. *(**Submodular Function Minimization**): A set $\hat{S}_{\text{sfm}}$ with $f(\hat{S}_{\text{sfm}}) \leq f(S_{\text{sfm}}^*) + 2n\varepsilon$.*

2. *(**Unrestricted Sparsest Submodular Set**): A set $\hat{S}_{\text{sparse}}$ with $\frac{f(\hat{S}_{\text{sparse}})}{|\hat{S}_{\text{sparse}}|} \leq \frac{f(S_{\text{sparse}}^*)}{|S_{\text{sparse}}^*|} + 2\varepsilon$.*

3. *(**Unrestricted Densest Supermodular Set**): A set $\hat{S}_{\text{dense}}$ with $\frac{f(\hat{S}_{\text{dense}})}{|\hat{S}_{\text{dense}}|} \geq \frac{f(S_{\text{dense}}^*)}{|S_{\text{dense}}^*|} - 2\varepsilon$.*

Notably, this theorem shows that approximating SFM, DSS, UDSS, or USSS reduces to approximating the minimum norm point in $B(f)$.

*Remark* 14. The condition in Theorem 13 assumes an approximate solution $\hat{x}$ satisfying $\|\hat{x}\|_2^2 \leq \langle q, \hat{x} \rangle + \varepsilon^2$ for all $q \in B(f)$, which upper bounds the duality gap for MNP. While Theorem 13 focuses on how this approximate leads to good set-based solutions for other problems (like SFM, USSS), it is worth remarking that that this same $\hat{x}$ is also an approximate solution to the MNP problem itself, in the sense that is close to the true minimum-norm point up to additive $O(\epsilon)$ error.

## 3 SUPERGREEDY++ and Fujishige-Wolfe's FW-MNP Algorithm as Universal Solvers

Let $f : 2^V \to \mathbb{R}$ be a normalized submodular or supermodular function. As indicated in Theorem 13, the main challenge in obtaining approximations for USSS, DSS, UDSS, and SFM is to compute an approximate minimum-norm-point $\hat{x} \in B(f)$. We will discuss three different methods, which we will refer to collectively as *Universal Solvers*.

**Frank-Wolfe Algorithm.** A classical method for solving MNP is the Frank-Wolfe algorithm, an iterative approach for minimizing a convex function $h : \mathcal{D} \to \mathbb{R}$ over a compact convex set $\mathcal{D}$. Each iteration computes the *linear minimization oracle* (LMO) $d^{(k)} = \arg\min_{s \in \mathcal{D}} \langle s, \nabla h(x^{(k-1)}) \rangle$, and updates $x^{(k)} = (1 - \alpha_k) x^{(k-1)} + \alpha_k d^{(k)}$ for $\alpha_k = \frac{2}{k+2}$. For MNP, we set $\mathcal{D} = B(f)$ and $h(x) = \|x\|_2^2$, reducing the LMO to $d^{(k)} = \arg\min_{d \in B(f)} \langle d, x^{(k-1)} \rangle$. This oracle is efficiently realized via Edmonds' greedy algorithm for (super)submodular functions[22].

**Fujishige-Wolfe Minimum Norm Point Algorithm.** Another standard approach is the Fujishige-Wolfe algorithm [27], which computes $\hat{x} \in B(f)$ satisfying $\|\hat{x}\|_2^2 \leq \langle \hat{x}, q \rangle + \varepsilon^2$ for all $q \in B(f)$. The algorithm requires $O(|V|Q^2/\varepsilon^2)$ iterations, where $Q = \max_{q \in B(f)} \|q\|_2$.

Like Frank-Wolfe, the Fujishige-Wolfe FW-MNP algorithm repeatedly calls the same linear minimization oracle over $B(f)$ but additionally requires an oracle that minimizes $\|x\|_2^2$ over the affine

hull of a set $S$ of selected extreme points of $B(f)$. This affine minimization oracle can be implemented efficiently in $O(|S|^3 + n|S|^2)$ time by computing the inverse of $B^\top B$, where $B$ is the $n \times |S|$ matrix whose columns are the points in $S$.

**SUPERGREEDY++.** Next, we demonstrate that SUPERGREEDY++, originally designed for solving the DSS problem, can be used to approximate MNP. Consequently, it is a universal solver for approximating the problems mentioned above.

**Theorem 15.** *Given a normalized submodular or supermodular function $f : 2^V \to \mathbb{R}$, SUPER-GREEDY++ returns a vector $\hat{x}$ satisfying $\|\hat{x}\|_2^2 \leq \langle q, \hat{x} \rangle + \varepsilon^2$ for all $q \in B(f)$, after*

$$
T = \tilde{O}\left( \frac{\max\limits_{s,d \in B(f)} \|s - d\|_2^2 + n \sum_{u \in V} f(u \mid V - u)^2}{\varepsilon^2} \right)
$$

*iterations, where $\tilde{O}$ hides polylogarithmic factors, and $f(u \mid V - u) = f(V) - f(V \setminus \{u\})$. Consequently, all approximation guarantees from Theorem 13 follow after the same number of iterations.*

*Proof in Appendix A.4..*

*Remark* 16. It is worth emphasizing that although SUPERGREEDY++ demonstrates outstanding *empirical* performance, its *theoretical* runtime remains pseudo-polynomial due to the presence of function-dependent terms. This contrasts with state-of-the-art SFM algorithms, which achieve strongly polynomial runtimes. Consequently, applying SUPERGREEDY++ directly to an SFM instance under our analysis results in a pseudo-polynomial algorithm rather than a strongly polynomial one.

## 4 Experiments

We evaluate our algorithms and baselines on a diverse set of datasets and problem settings, running over 400 experiments across SFM, USSS, and UDSS. Our experiments aim to answer three key questions:

- Which algorithms perform best for which problems, especially when applied beyond their original setting? Are there cases where general-purpose methods (e.g., FW-MNP for USSS) outperform problem-specific heuristics?
- How do convex optimization methods (e.g., Frank-Wolfe, Fujishige-Wolfe FW-MNP) compare to combinatorial (e.g. SUPERGREEDY++), LP-based, and flow-based approaches in runtime and solution quality?
- Can SUPERGREEDY++, Frank-Wolfe, and FW-MNP serve as effective general-purpose solvers across multiple problem classes?

Our empirical study is organized *per problem class*, covering Unrestricted Sparsest Submodular Set (USSS), Submodular Function Minimization (SFM), and Unrestricted Densest Supermodular Set (USSS). We specify problems, datasets, algorithms, and evaluation metrics for each. Experiments were run in parallel on a Slurm-managed cluster (AMD EPYC 7763, 128 cores, 256GB RAM). All methods are implemented in C++20[2], primarily by the authors, except for specialized baselines (e.g., GREEDY++). **Due to space constraints, we only discuss two experiments here; most of our experimental results are provided in the appendix.** Specifically, we selected these two experiments because the HNSN results demonstrate the value of classic algorithms for a recently explored problem, whereas the Minimum $s - t$ Cut results illustrate the value of newly developed algorithms for a long-standing problem.

**(1) Heavy Nodes in a Small Neighborhood (HNSN).** Recall the HNSN problem from the introduction. Since the function $f(S) = |N(S)|$ is submodular, HNSN is a special case of the weighted USSS problem. Ling *et al.* further reformulate the problem over the left vertex set $L$, defining the function $\overline{N} : 2^L \to \mathbb{R}$ by $\overline{N}(S) = \{v \in R \mid \delta(v) \subseteq S\}$ where $\delta(v)$ denotes the set of neighbors of $v$. Under this formulation, the problem becomes that of maximizing $w(\overline{N}(S))/|S|$ over non-empty subsets $S \subseteq L$. We adopt this viewpoint in our implementation.

---

[2]Our code and datasets are available at https://github.com/FaroukY/CorporateEquivalence

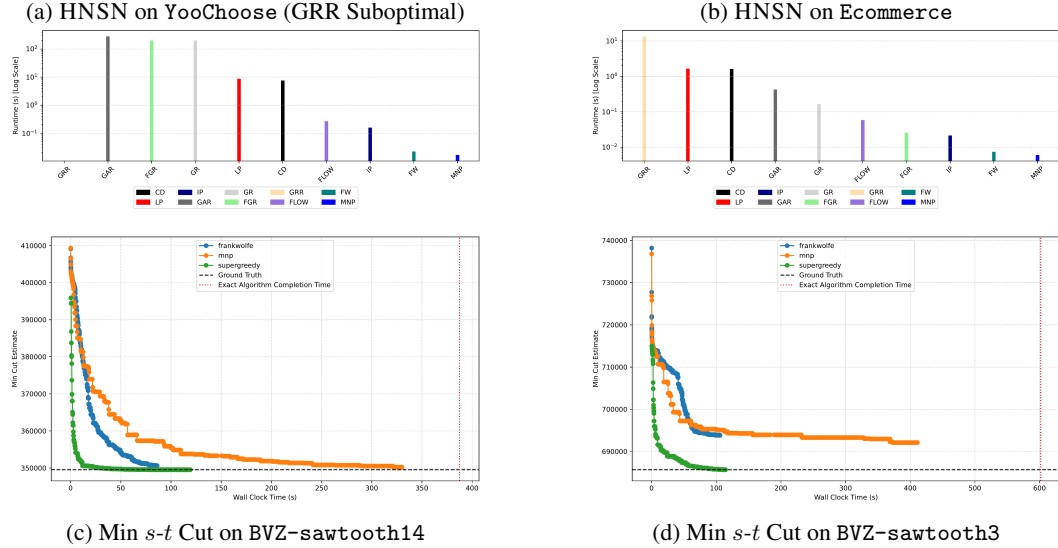

Figure 1: Performance of algorithms across selected HNSN and Minimum $s$-$t$ Cut instances.

**Algorithms.** We compare against the six baselines of Ling *et al.* [42], and introduce a novel flow-based algorithm (detailed in Appendix B). In total, we evaluate **ten algorithms**: IP (SUPERGREEDY++), FW (Frank-Wolfe), FW-MNP (Fujishige-Wolfe Minimum Norm Point Algorithm), FLOW (our new flow-based method), CD (ContractDecompose, [42]), LP (Linear Programming, solved via Gurobi with academic license), GAR (Greedy Approximation, [42]), GR (Greedy, [42]), FGR (Fast Greedy, [42]), and GRR (GreedRatio, [5]). The first four are our implementations; the remaining six follow Ling *et al.*'s code [42].

**Datasets.** Table 1 summarizes the datasets used for the HNSN problem, following the benchmarks introduced by Ling *et al.* [42]. Each dataset is represented as a weighted bipartite graph with left vertices $L$, right vertices $R$, weights on the right vertices $w : R \to \mathbb{R}_{\geq 0}$, and edges $E$. The graphs are derived from diverse real-world domains, including recommendation systems (e.g., YooChoose, Kosarak), citation networks (ACM), social networks (NotreDame, IMDB, Digg), and financial transactions (e.g., E-commerce, Liquor). The datasets exhibit a wide range of sizes, with up to $\sim 10^6$ vertices and $\sim 2 \times 10^7$ edges.

**Discussion of HNSN Results.** All algorithms were given a 30-minute time limit per dataset. The full results are provided in Appendix C; we plot two example runs in Figures 1a and 1b. Across all datasets, FW and FW-MNP consistently emerged as the fastest to converge. In nearly every case, FW-MNP outperformed FW, achieving the best solution quality in the shortest time, with only a single dataset where FW was slightly faster. Relative to the six baselines from Ling *et al.* [42], these methods achieved speedups ranging from $5\times$ to $595\times$. Figures 1a, 1b show two examples, with the rest in the appendix.

Our new flow-based algorithm and SUPERGREEDY++ (IP) typically followed, ranking third and fourth in performance across datasets. Figure 1a shows a particularly striking example where, on the YooChoose dataset, FW-MNP achieved a $595\times$ speedup over the best previously published baseline. In this instance, all algorithms converged to the correct final density, except for GRR, which terminated at a suboptimal solution and was therefore excluded from the runtime comparison.

These results underscore a recurring insight consistently observed across our experiments: universal solvers such as FW-MNP and FW, originally developed for broader optimization tasks, can significantly outperform specialized heuristics when applied to specific problems like HNSN, provided they are used within the proper conceptual framework that we have outlined. The results also show the relevance of classic optimization-based methods for new problems.

**(2) Minimum $s$-$t$ Cut.** We next evaluate our algorithms on the classical *Minimum $s$-$t$ Cut* problem, a fundamental task in combinatorial optimization with widespread applications. Specifically,

the Minimum $s$-$t$ Cut problem can be framed as minimizing the normalized submodular function $g(S) = |\delta(S \cup \{s\})| - |\delta(\{s\})|$ over subsets $S \subseteq V \setminus \{s, t\}$, ensuring that any feasible solution corresponds to a valid $s$-$t$ cut. Within this formulation, the Minimum $s$-$t$ Cut problem becomes an instance of *Submodular Function Minimization (SFM)*, and thus naturally fits within the unified framework established in this paper. This connection allows us to apply universal solvers, such as SUPERGREEDY++, Frank-Wolfe, and Wolfe's Minimum Norm Point algorithm, to this classical combinatorial problem.

**Algorithms.** We evaluate three iterative algorithms—Frank-Wolfe (FRANKWOLFE), Fujishige-Wolfe Minimum Norm Point (FW-MNP), and SUPERGREEDY++ (SUPERGREEDY)—on the Minimum $s$-$t$ Cut problem. We also compute the exact minimum cut value for each instance using standard combinatorial flow-based methods (Edmonds Karp algorithm), serving as ground truth. This allows us to assess both the solution quality and runtime efficiency of the optimization-based approaches relative to the exact combinatorial solution.

**Datasets.** Our evaluation spans two groups of datasets. First, we consider four classical benchmark instances from the first DIMACS Implementation Challenge on minimum cut [21], known for their small size but challenging cut structures. Second, to assess scalability and robustness, we include a large-scale dataset family, specifically the BVZ-SAWTOOTH instances [38], which consist of 20 distinct minimum $s$-$t$ cut problems. Each instance contains approximately 500,000 vertices and 800,000 edges, testing algorithmic scalability.

**Discussion of Minimum $s$-$t$ Cut Results.** Full results are provided in Appendix D; we plot two select examples in Figures 1c and 1d. Across all datasets, SUPERGREEDY++ was consistently the fastest to approximate the minimum cut, often by large margins over exact flow-based methods, FW-MNP, and FW. In 16 of 24 instances, it found the exact min-cut rapidly. In the remaining cases, its solution was within $1.000023\times$ the optimum after 500 iterations at most, always converging before the exact flow solver. A notable example is the BVZ-SAWTOOTH14 instance, where SUPERGREEDY++ converges orders of magnitude faster (Figure 1c). This strong performance parallels its success in dense subgraph problems, where it efficiently finds high-quality solutions but can slow down near the optimum. Combining SUPERGREEDY++ with flow-based refinement remains an interesting direction for future work. *Interestingly, while* FW-MNP *and* FW *excel on* HNSN*, they both underperform here, underscoring how problem structure affects solver efficiency.* Finally, the experiments show that recent heuristics like SUPERGREEDY++ can compete with classical max-flow algorithms for the minimum $s - t$ cut problem.

**Additional Experiments.** Beyond the two highlighted experiments, we conduct a comprehensive evaluation across additional problem instances corresponding to each of the general problem classes studied. For SFM, we include the classical Contrapolymatroid Membership problem (Appendix H), which tests whether a given vector belongs to the base polytope. For DSS, we evaluate both the Densest Subgraph problem (Appendix E) and the generalized $p$-mean Densest Subgraph problem [58] (Appendix F). For UDSS, we consider the Anchored Densest Subgraph problem (Appendix G). We also report experiments on the MNP problem (Appendix I). We refer readers to the appendix for the complete set of experiments across all problem classes.

**Flow-Based Methods: Strengths and Limitations.** For the classical Densest Subgraph problem, our experiments reaffirm that flow-based methods remain among the fastest approaches when combined with the density-improvement framework introduced by Huang *et al.* [34] and Hochbaum [33]. While Hochbaum uses the PseudoFlow algorithm for this task, our experiments demonstrate that comparable performance can be achieved using more standard push-relabel max-flow solvers within the same density-improvement framework. Essentially, the primary source of speedup comes from the iterative density-improvement strategy rather than the specific choice of flow solver. This holds because for a fixed density threshold $\lambda$, finding a subset $S$ minimizing $\lambda|S| - f(S) = \lambda|S| - |E(S)|$ naturally reduces to a standard min-cut instance. A similar phenomenon occurs in the HNSN problem: as we outline in the appendix, there exists a flow reduction that efficiently solves subproblems of the form $\lambda|S| - f(S)$ for HNSN, making flow-based methods highly effective here as well.

However, flow-based methods are not universally applicable. For more general problems, such as the generalized $p$-mean Densest Subgraph problem, no known linear flow network formulations can minimize objectives like $\lambda|S| - f(S)$ for arbitrary supermodular functions. In such cases, flow-based methods cannot be used, and one must resort to algorithms like SUPERGREEDY++, Frank-

Table 1: Summary of HNSN Datasets. All graphs are weighted bipartite graphs.

| Dataset | $|L|$ | $|R|$ | $|E|$ | $k$ (Connected components) |
|---|---|---|---|---|
| Foodmart (FM) | 1,559 | 4,141 | 18,319 | 1 |
| E-commerce (EC) | 3,468 | 14,975 | 174,354 | 12 |
| Liquor (LI) | 4,026 | 52,131 | 410,609 | 165 |
| Fruithut (FR) | 1,265 | 181,970 | 652,773 | 4 |
| YooChoose (YC) | 107,276 | 234,300 | 507,266 | 22,033 |
| Kosarak (KS) | 41,270 | 990,002 | 8,019,015 | 271 |
| Connectious (CN) | 458 | 394,707 | 1,127,525 | 117 |
| Digg (DI) | 12,471 | 872,622 | 22,624,727 | 13 |
| NotreDame (ND) | 127,823 | 383,640 | 1,470,404 | 3,142 |
| IMDB (IM) | 303,617 | 896,302 | 3,782,463 | 7,885 |
| NBA Shot (NBA) | 129 | 1,603 | 13,726 | 1 |
| ACM Citation (ACM) | 751,407 | 739,969 | 2,265,837 | 1 |

Wolfe, or FW-MNP, which only require access to a value oracle. These methods thus offer broader applicability across diverse problem settings where flow reductions are unavailable.

## 5   Conclusion and Limitations

The overarching conclusion of our study is clear: SUPERGREEDY++, Frank-Wolfe, and the Fujishige-Wolfe FW-MNP algorithms consistently achieve state-of-the-art performance across a wide range of submodular and supermodular ratio problems. Given the broad applicability of DSS, USSS, SFM, and related formulations, we advocate for these three methods to be included as essential baselines in future empirical evaluations of such problems.

A limitation of our current work is that while one of these algorithms consistently outperforms problem-specific heuristics, the identity of the best-performing method varies with the problem instance. Developing a deeper understanding of when and why each algorithm excels remains an open question. Moreover, while our results provide general additive approximation guarantees via reductions to the Minimum Norm Point problem, it remains an interesting open question to develop a direct, problem-specific analysis of algorithms like SUPERGREEDY++ in settings such as minimum $s$-$t$ cut. Finally, while we discussed related work on coordinate descent methods, such as recennt work by Nguyen *et al.* [47], we did not dive deep into the broader family of decomposable submodular function minimization (DSFM) techniques in detail. A systematic exploration of general DSFM methods and their applicability remains an important direction for future research.

Additionally, while our paper focuses on **unrestricted** versions to allow non-monotone and negative functions (which are common in real-world applications like ADS and HNSN), restricted versions such as Sparsest Submodular Set with Monotone or Positive $f$ could offer both: 1) Simpler theoretical analysis, and 2) Opportunities for improved approximations or faster algorithms. It is also worth noting that for DSS, one can obtain results in terms of multiplicative approximations that cannot be shown for UDSS [17]. We have some preliminary results for SSS in similar vein, for example SuperGreedy performance in terms of curvature constant of $f$ or potentially convergence analysis for SuperGreedy++ that is similar to that for DSS in [17] for a multiplicative -approximation rather than additive one.

In many problem-specific heuristics (e.g., greedy peeling or local ratio methods), the "specific structure" that is used is often simplistic or myopic, such as removing low-degree nodes without a global view. In contrast, general-purpose methods like FW-MNP, Frank-Wolfe, and SUPERGREEDY++ implicitly operate on the full combinatorial or polyhedral structure of the problem. As a result, they can often exploit hidden structure better than naive heuristics.

That said, **it remains possible to design better problem-specific algorithms that leverage deeper properties** (e.g., flow-augmenting paths, dual decompositions). Our results highlight that some heuristics for specific problems underperform not because the problem is hard, but because the heuristic doesn't fully exploit the structure, which we believe is an exciting direction for future algorithm design in those problems.

# 6 Acknowledgment

This work used Delta CPU at NCSA through allocation CIS250079 from the ACCESS [8] program, which is supported by U.S. National Science Foundation grants #2138259, #2138286, #2138307, #2137603, and #2138296.

Chandra Chekuri and Elfarouk Harb are supported in part by NSF grant CCF-2402667.

Yousef Yassin gratefully acknowledges support from the Ontario Graduate Scholarship, and the Vector Scholarship in AI from the Vector Institute.

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

Table 2: Acronyms and abbreviations used throughout the paper.

| Acronym | Full name | Brief description / where used |
|---|---|---|
| *Problem classes* | | |
| **SFM** | Submodular Function Minimization | $\min_{\emptyset \neq S \subseteq V} f(S)$ for submodular $f$. |
| **DSG** | Densest Subgraph | $\max_{S \neq \emptyset} \frac{|E(S)|}{|S|}$. Special case of DSS. |
| **DSS** | Densest Supermodular Set | $\max_{S \neq \emptyset} \frac{f(S)}{|S|}$ for normalized, monotone supermodular $f$. |
| **USSS** | Unrestricted Sparsest Submodular Set | $\min_{S \neq \emptyset} \frac{f(S)}{|S|}$ for (possibly non-monotone/negative) submodular $f$. |
| **UDSS** | Unrestricted Densest Supermodular Set | $\max_{S \neq \emptyset} \frac{f(S)}{|S|}$ for (possibly non-monotone/negative) supermodular $f$. |
| **MNP** | Minimum Norm Point | $\min_{x \in B(f)} \|x\|_2^2$ over the base polytope $B(f)$. |
| **ADS** | Anchored Densest Subgraph | Density with vertex penalties outside anchor set $R$. |
| **HNSN** | Heavy Nodes in a Small Neighborhood | Bipartite ratio objective: maximize $w(\overline{N}(S))/|S|$. |
| *Algorithms / optimization terms* | | |
| **FW** | Frank–Wolfe (a.k.a. conditional gradient) | First-order method used to solve MNP over $B(f)$. |
| **FW-MNP** | Fujishige–Wolfe MNP algorithm | Specialized algorithm for MNP on submodular/supermodular base polytopes. |
| **FISTA** | Fast Iterative Shrinkage–Thresholding Algorithm | Accelerated first-order method referenced in related work. |
| **LMO** | Linear Minimization Oracle | Oracle $\arg\min_{s \in \mathcal{D}} \langle s, \nabla h(x) \rangle$ used by FW-type methods. |
| **LP** | Linear Programming | Used as a baseline/alternative exact method. |
| **IP** | Iterative Peeling (SuperGreedy++) | Name used in experiments for SUPER-GREEDY++. |
| *Benchmarks / misc.* | | |
| **DIMACS** | Center for Discrete Mathematics and Theoretical Computer Science | Classic min-cut challenge benchmark suite. |

# A    Proofs

## A.1    Proof of Lemma 11

*Proof.* Fix $\lambda$ and $S \subseteq V$. Since $x^* \in B(f)$, we have

$$f(S) - \lambda|S| \geq \sum_{v \in S}(x_v^* - \lambda). \tag{1}$$

The right-hand side of (1) is minimized by taking $S_\lambda = \{v \in V \mid x_v^* \leq \lambda\}$, so

$$\sum_{v \in S}(x_v^* - \lambda) \geq \sum_{v \in S_\lambda}(x_v^* - \lambda) = x^*(S_\lambda) - \lambda|S_\lambda|. \tag{2}$$

We claim that $x^*(S_\lambda) = f(S_\lambda)$. Combining this with (1) and (2) gives

$$f(S_\lambda) - \lambda|S_\lambda| \leq f(S) - \lambda|S|,$$

as desired.

To prove the claim, order $V$ so that $x_{v_1}^* \leq x_{v_2}^* \leq \cdots \leq x_{v_n}^*$, and define $S_i = \{v_1, \ldots, v_i\}$. For some $i$, we have $S_\lambda = S_i$, so it suffices to show $x^*(S_i) = f(S_i)$ for all $i$. This follows from [26] Lemma 7.4 (pp. 218–219, equations 7.12–7.15).

$\square$

## A.2    Proof of Theorem 12

*Proof.* **(1) $\Rightarrow$ (4)/(5):** Let $f : 2^V \to \mathbb{R}$ be a normalized submodular or supermodular function. We focus on submodular functions for USSS; the supermodular case (UDSS) is analogous via negation.

Our goal is to compute the minimum norm point $x^* \in B(f)$ using an oracle for USSS. We iteratively build $x^*$ by decomposing $f$ into a sequence of sparsest sets.

Initialize $f_0 = f$. At iteration $i \geq 1$, define $S_i$ as the minimizer of $f_i(S)/|S|$ over non-empty $S \subseteq V \setminus (S_1 \cup \cdots \cup S_{i-1})$. Set $x_v = f_i(S_i)/|S_i|$ for all $v \in S_i$.

After assigning $x_v$ for $v \in S_i$, contract $S_i$ by defining:

$$f_{i+1}(S) = f_i(S \cup S_i) - f_i(S_i), \quad \forall S \subseteq V \setminus (S_1 \cup \cdots \cup S_i).$$

Repeat until $S_1 \cup \cdots \cup S_k = V$. This process requires at most $n$ iterations since the $S_i$ are disjoint.

**Claim:** The resulting vector $x$ is the minimum norm point in $B(f)$.

*Proof of Claim:* Let $x$ be the vector constructed by the above process, where for each $v \in S_i$, we set $x_v = \lambda_i = f_i(S_i)/|S_i|$.

We now prove that $x$ is the minimum norm point in $B(f)$ by showing that $x$ is the lexicographically maximum base of $B(f)$. Recall that the lex-maximal vector corresponds to the minimum norm point in a polymatroid.

We proceed by induction on $i$ to show that for each $i$, any lex-minimal base $x^* \in B(f)$ must satisfy $x_v^* = \lambda_i$ for all $v \in S_i$.

Base case ($i = 1$): Since $x^* \in B(f)$, we have

$$x^*(S_1) \leq f(S_1) = \lambda_1|S_1|.$$

But by construction, $x_v = \lambda_1$ for all $v \in S_1$, so $x(S_1) = \lambda_1|S_1|$.

Therefore, $x^*(S_1) \leq x(S_1)$. In particular, there exists at least one $v \in S_1$ with $x_v^* \leq \lambda_1$.

Since $x^*$ is lexicographically maximal, we must have $x_v^* = \lambda_1$ for all $v \in S_1$; otherwise, if any $x_v^* < \lambda_1$ for $v \in S_1$, lex-maximality would be violated.

Induction step: Assume for $j = 1, \ldots, i$, we have $x_v^* = \lambda_j$ for all $v \in S_j$.

Consider $S_{i+1}$. Since $x^* \in B(f)$, we have

$$x^*(S_1 \cup \cdots \cup S_{i+1}) \leq f(S_1 \cup \cdots \cup S_{i+1}).$$

Using telescoping sums, this becomes:

$$f(S_1 \cup \cdots \cup S_{i+1}) = \sum_{h=1}^{i+1} [f(S_1 \cup \cdots \cup S_h) - f(S_1 \cup \cdots \cup S_{h-1})] = \sum_{h=1}^{i+1} \lambda_h |S_h|.$$

Thus,

$$x^*(S_{i+1}) = x^*(S_1 \cup \cdots \cup S_{i+1}) - x^*(S_1 \cup \cdots \cup S_i) \leq \lambda_{i+1} |S_{i+1}|,$$

where the induction hypothesis gives $x^*(S_1 \cup \cdots \cup S_i) = \sum_{h=1}^{i} \lambda_h |S_h|$.

Hence, the average value of $x_v^*$ over $v \in S_{i+1}$ is at most $\lambda_{i+1}$. Therefore, there must exist at least one $v \in S_{i+1}$ with $x_v^* \leq \lambda_{i+1}$.

Since $x^*$ is lexicographically maximal, we must have $x_v^* = \lambda_{i+1}$ for all $v \in S_{i+1}$; otherwise, if any $x_v^* < \lambda_{i+1}$, lex-maximality would be contradicted.

By induction, $x_v^* = \lambda_i$ for all $v \in S_i$ and all $i$. Hence, $x^* = x$. Thus, the constructed vector $x$ is the lex-maximal base of $B(f)$, which is also the minimum norm point.

**(4) $\Leftrightarrow$ (5):** Immediate from duality. If $f$ is supermodular, $-f$ is submodular. The sparsest submodular set problem becomes densest supermodular set under negation. Hence, solving USSS is equivalent to solving UDSS on $-f$.

**(5) $\Rightarrow$ (3):** Given a normalized supermodular function $f$, define:

$$C = \max\{0, \max_{S \subseteq V}(-f(S))\},$$

and set $g(S) = f(S) + C|S|$. Then $g$ is normalized, monotone, non-negative, and supermodular. Furthermore,

$$\arg \max_{S \subseteq V} \frac{g(S)}{|S|} = \arg \max_{S \subseteq V} \frac{f(S)}{|S|},$$

as adding a constant $C$ to all sets gains affects neither the maximizer (as it just shifts the sparsity ratio). Thus, UDSS reduces to DSS.

**(3) $\Rightarrow$ (2):** This reduction is classical (e.g., [20, 34, 33]). Given monotone, normalized, non-negative supermodular $f$, we iteratively refine a maximizer of $f(S)/|S|$ using SFM. Initialize $S_0 = V$, $\lambda_1 = f(S_0)/|S_0|$. At iteration $i$, solve:

$$S_i = \arg \min_{S \subseteq V, S \neq \emptyset} (\lambda_i |S| - f(S)),$$

using SFM. Update $\lambda_{i+1} = f(S_i)/|S_i|$. Iterate until $\lambda_k |S_k| - f(S_k) = 0$. Since the sequence of $\lambda_i$ decreases and $f$ is normalized, the process terminates in at most $n$ steps. The last $S_{k-1}$ maximizes $f(S)/|S|$, solving DSS via SFM.

**(2) $\Rightarrow$ (1):** Given $f$, compute the minimum-norm point $x^* \in B(f)$. By Lemma 11, the set

$$S_0 = \{v \in V \mid x_v^* \leq 0\}$$

minimizes $f(S)$. Thus, SFM can be reduced to computing $x^*$, completing the reduction.

$\square$

### A.3 Proof of Theorem 13

*Proof.* **(1)** This result follows directly from the analysis in [11].

**(2)** We generalize the argument of [11]. Without loss of generality, reorder indices so that $\hat{x}_1 \leq \hat{x}_2 \leq \cdots \leq \hat{x}_n$. Consider the set $\hat{S} = [i] := \{1, \ldots, i\}$ that minimizes $f([i])/i$ over $i = 1, \ldots, n$.

Let $\lambda^*$ denote the density of the sparsest submodular set, i.e., $\lambda^* = \min_{S \neq \emptyset} f(S)/|S|$. Define $k$ as the smallest index satisfying:

(C1) $\hat{x}_{k+1} > \lambda^*$,

(C2) $\hat{x}_{k+1} - \hat{x}_k \geq \frac{\varepsilon}{k}$.

From [11], we have the key inequality:
$$\sum_{i=1}^{n-1}(\hat{x}_{i+1} - \hat{x}_i)(f([i]) - \hat{x}([i])) \leq \varepsilon^2.$$

Let $t = \big|\{i \in [k] : \hat{x}_i > \lambda^*\}\big|$. Observe:
$$\sum_{i\in[k]:\hat{x}_i>\lambda^*} \hat{x}_i = \lambda^* t + \sum_{i\in[k]:\hat{x}_i>\lambda^*} (\hat{x}_i - \lambda^*)$$
$$\leq \lambda^* t + \sum_{i\in[k]} \frac{i \cdot \varepsilon}{k}$$
$$\leq \lambda^* t + k\varepsilon,$$

where the second inequality follows since (C2) fails for all $i < k$ with $\hat{x}_i > \lambda^*$.

Since $\hat{x}_{k+1} - \hat{x}_k \geq \frac{\varepsilon}{k}$, it follows that
$$f([k]) - \hat{x}([k]) \leq k\varepsilon,$$

and therefore,
$$\frac{f([k])}{k} \leq \frac{\hat{x}([k]) + k\varepsilon}{k}$$
$$= \frac{\sum_{i\in[k]:\hat{x}_i\leq\lambda^*} \hat{x}_i + \sum_{i\in[k]:\hat{x}_i>\lambda^*} \hat{x}_i + k\varepsilon}{k}$$
$$\leq \frac{\lambda^*(k-t) + \lambda^* t + 2k\varepsilon}{k} = \lambda^* + 2\varepsilon.$$

**(3)** The argument parallels that of (2). We now reorder indices so that $\hat{x}_1 \geq \hat{x}_2 \geq \cdots \geq \hat{x}_n$, and select the set $\hat{S} = [i] := \{1, \ldots, i\}$ that maximizes $f([i])/i$.

Let $\lambda^* = \max_{S\neq\emptyset} f(S)/|S|$. Define $k$ as the smallest index satisfying:

(C1) $\hat{x}_{k+1} < \lambda^*$,

(C2) $\hat{x}_k - \hat{x}_{k+1} \geq \frac{\varepsilon}{k}$.

Using the analogous inequality from [11]:
$$\sum_{i=1}^{n-1}(\hat{x}_i - \hat{x}_{i+1})(\hat{x}([i]) - f([i])) \leq \varepsilon^2.$$

Let $t = \big|\{i \in [k] : \hat{x}_i < \lambda^*\}\big|$. We have:
$$\sum_{i\in[k]:\hat{x}_i<\lambda^*} \hat{x}_i = \lambda^* t + \sum_{i\in[k]:\hat{x}_i<\lambda^*} (\hat{x}_i - \lambda^*)$$
$$\geq \lambda^* t - \sum_{i\in[k]} \frac{i \cdot \varepsilon}{k}$$
$$\geq \lambda^* t - k\varepsilon,$$

where the second inequality holds since (C2) fails for all $i < k$ with $\hat{x}_i < \lambda^*$.

Since $\hat{x}_{k-1} - \hat{x}_k \geq \frac{\varepsilon}{k}$, we deduce:
$$\hat{x}([k]) - f([k]) \leq k\varepsilon,$$

and consequently,
$$\frac{f([k])}{k} \geq \frac{\hat{x}([k]) - k\varepsilon}{k}$$
$$= \frac{\sum_{i\in[k]:\hat{x}_i\geq\lambda^*} \hat{x}_i + \sum_{i\in[k]:\hat{x}_i<\lambda^*} \hat{x}_i - k\varepsilon}{k}$$
$$\geq \frac{\lambda^*(k-t) + \lambda^* t - 2k\varepsilon}{k} = \lambda^* - 2\varepsilon.$$

$\square$

## A.4 Proof of Theorem 15

*Proof.* Our analysis follows the approach of Harb *et al.* [32], based on the Frank-Wolfe framework of Jaggi [36].

**Jaggi's Frank-Wolfe Framework.** The Frank-Wolfe algorithm is a classical method for minimizing a convex function over a convex set. While its convergence has been known since [24], Jaggi's analysis [36] offers a modern, simplified treatment that expresses convergence rates in terms of the so-called *curvature constant* of the objective function.

**Definition 17** (Curvature constant). Let $\mathcal{D} \subseteq \mathbb{R}^d$ be a compact convex set, and $h : \mathcal{D} \to \mathbb{R}$ a convex, differentiable function. The curvature constant $C_h$ is defined as

$$C_h = \sup_{\substack{x, s \in \mathcal{D}, \, \gamma \in [0,1] \\ y = x + \gamma(s-x)}} \frac{2}{\gamma^2} \left( h(y) - h(x) - \langle y - x, \nabla h(x) \rangle \right).$$

One advantage of Jaggi's formulation is that it seamlessly extends to *approximate* versions of Frank-Wolfe, where the linear minimization oracle (LMO) may only be computed approximately.

**Definition 18** (Approximate LMO). Given $h : \mathcal{D} \to \mathbb{R}$, an $\varepsilon$-approximate linear minimization oracle returns $\hat{s} \in \mathcal{D}$ satisfying

$$\langle \hat{s}, \nabla h(w) \rangle \leq \langle s^*, \nabla h(w) \rangle + \varepsilon,$$

where $s^* = \arg\min_{s \in \mathcal{D}} \langle s, \nabla h(w) \rangle$ is the exact LMO.

Jaggi's main result shows that if, at iteration $k$, we compute an approximate LMO with error at most $\frac{\delta C_h}{k+2}$, then the Frank-Wolfe algorithm converges with only a constant-factor slowdown.

**Lemma 19** ([36]). *Define the* duality gap *at $x \in \mathcal{D}$ as*

$$g(x) = \max_{q \in \mathcal{D}} \langle x - q, \nabla h(x) \rangle.$$

*If we use a $\frac{\delta C_h}{k+2}$-approximate LMO at iteration $k$, then after $K$ iterations there exists an iterate $x^{(k)}$, with $1 \leq k \leq K$, satisfying*

$$g(x^{(k)}) \leq \frac{7 C_h}{K + 2}(1 + \delta).$$

**Applying to SUPERGREEDY++ and FW-MNP.** Our goal is to analyze SUPERGREEDY++ for the Minimum Norm Point (MNP) problem on the base polytope of a normalized submodular or supermodular function $f : 2^V \to \mathbb{R}$. Specifically, we consider

$$h(x) = \|x\|_2^2, \quad \mathcal{D} = B(f).$$

For this quadratic objective, the curvature constant admits a simple form:

**Lemma 20.**

$$C_h = 2 \max_{s, d \in B(f)} \|s - d\|_2^2.$$

*Proof.* This follows by substituting $h(x) = \|x\|_2^2$ into the definition of $C_h$. The supremum is attained when $x$, $s$, and $d$ are chosen to maximize $\|s - d\|_2^2$, yielding the claimed expression. $\square$

Next, note that we can, without loss of generality, assume $f$ is a supermodular function. This is because submodular and supermodular MNP problems are duals of each other under negation:

**Lemma 21.** *Given a normalized submodular function $g$, define $f = -g$. If we can solve the MNP problem for $f$, i.e., find $\hat{x} \in B(f)$ such that*

$$\|\hat{x}\|_2^2 \leq \langle q, \hat{x} \rangle + \varepsilon^2 \quad \text{for all } q \in B(f),$$

*then we can obtain a corresponding solution $x' = -\hat{x} \in B(g)$ with the same guarantee:*

$$\|x'\|_2^2 \leq \langle q, x' \rangle + \varepsilon^2 \quad \text{for all } q \in B(g).$$

*Proof.* Given $g$, let $f = -g$. Since $f$ is submodular, compute $\hat{x} \in B(f)$ such that

$$\|\hat{x}\|_2^2 \leq \langle q, \hat{x} \rangle + \varepsilon^2 \quad \text{for any } q \in B(f).$$

Define $x' = -\hat{x}$. Observe that $q \in B(f)$ if and only if $-q \in B(g)$. Specifically, $q(V) = f(V)$ implies $-q(V) = -f(V) = g(V)$, and $q(S) \geq f(S)$ if and only if $-q(S) \leq -f(S) = g(S)$. Thus, $x' \in B(g)$ and, for any $q \in B(g)$, since $-q, -x' \in B(f)$,

$$\|x'\|_2^2 = \|-x'\|_2^2 \leq \langle -q, -x' \rangle + \varepsilon^2 = \langle q, x' \rangle + \varepsilon^2.$$

$\square$

**Approximate LMO in SUPERGREEDY++.** Algorithm 2 reframes SUPERGREEDY++ as a "noisy" Frank-Wolfe algorithm. At each iteration, SUPERGREEDY++ computes a descent direction using the PEELWEIGHTED subroutine. While this is not an exact LMO, Harb *et al.* [32] showed that it approximates the LMO to within an error term that decays with $t$:

**Lemma 22** ([32]). *Let $s_t$ denote the exact LMO direction at iteration $t$, and $\hat{d}_t$ be the direction returned by* PEELWEIGHTED. *Then,*

$$\langle s_t, b^{(t-1)} \rangle \leq \langle \hat{d}_t, b^{(t-1)} \rangle + \frac{n \sum_{u \in V} f(u \mid V \setminus \{u\})^2}{t}.$$

This shows that at iteration $t$, SUPERGREEDY++ uses a

$$\frac{n \sum_{u \in V} f(u \mid V \setminus \{u\})^2}{t}$$

approximate LMO error. Normalizing by the curvature constant $C_h$, this corresponds to

$$\delta = \frac{n \sum_{u \in V} f(u \mid V \setminus \{u\})^2}{C_h}.$$

**Convergence of SUPERGREEDY++.** Substituting this approximation quality into Jaggi's convergence lemma, we find that after $T$ iterations, there exists an iterate $x^{(k)}$ with

$$g(x^{(k)}) \leq \frac{7C_h}{T+2} (1+\delta) = \frac{7C_h}{T+2} \left( 1 + \frac{n \sum_{u \in V} f(u \mid V \setminus \{u\})^2}{C_h} \right).$$

Simplifying, this gives

$$g(x^{(k)}) \leq \frac{7}{T+2} \left( C_h + n \sum_{u \in V} f(u \mid V \setminus \{u\})^2 \right).$$

To ensure $g(x^{(k)}) \leq \varepsilon^2$, it suffices to choose

$$T = O\left( \frac{C_h + n \sum_{u \in V} f(u \mid V \setminus \{u\})^2}{\varepsilon^2} \right) = O\left( \frac{\max_{s,d \in B(f)} \|s - d\|_2^2 + n \sum_{u \in V} f(u \mid V \setminus \{u\})^2}{\varepsilon^2} \right),$$

The $\tilde{O}$ polylog term comes from the fact that we use a learning rate (in SUPERGREEDY++) of $1/(t+1)$ instead of $2/(t+2)$.

$\square$

---

**Algorithm 1** PeelingWeighted++

---

1: **function** PEELWEIGHTED$\left(f : 2^V \to \mathbb{R},\ w \in \mathbb{R}\right)$
2: Initialize: $\hat{d}(u) = 0$ for all $u \in V$
3: $S_1 \leftarrow V$
4: **for** $j = 1$ to $n$ **do**
5:     $v_j \leftarrow \arg\max_{v \in S_j}\{w(v) + f(v \mid S_j - v)\}$
6:     $\hat{d}(v_j) \leftarrow f(v_j \mid S_j - v_j)$
7:     $S_{j+1} \leftarrow S_j \setminus \{v_j\}$
8: **end for**
9: **return** $\hat{d}$

---

---

**Algorithm 2** SuperGreedy++ For Minimum Norm Point

---

1: **function** SUPERGREEDY++$\left(f : 2^V \to \mathbb{R},\ T\right)$
2: Initialize: $x^{(0)}(u) = 0$ for all $u \in V$
3: **for** $t = 1$ to $T$ **do**
4:     $\hat{d}_t \leftarrow$ PEELWEIGHTED$(f, (t-1)x^{(t-1)})$
5:     $x^{(t)} \leftarrow \left(1 - \frac{1}{t+1}\right)x^{(t-1)} + \frac{1}{t+1}\hat{d}_t$
6: **end for**
7: **return** $x^{(k)}$, for $0 \leq k \leq T$, that minimizes $\|x^{(k)}\|_2^2 - \max_{q \in B(f)}\langle q, x^{(k)}\rangle$

---

# B  A New Flow-Based Algorithm for HNSN

Recall the HNSN problem from the introduction. Since the function $f(S) = |N(S)|$ is submodular, HNSN is a special case of the weighted USSS problem. Ling *et al.* further reformulate the problem over the left vertex set $L$, defining the function $\overline{N} : 2^L \to \mathbb{R}$ by $\overline{N}(S) = \{v \in R \mid \delta(v) \subseteq S\}$ where $\delta(v)$ denotes the set of neighbors of $v$. Under this formulation, the problem becomes that of maximizing $w(\overline{N}(S))/|S|$ over non-empty subsets $S \subseteq L$. We adopt this viewpoint in our implementation.

We adopt the density improvement framework from the proof of Theorem 12. We begin by setting $S_0 = L$ and define the function $f : 2^L \to \mathbb{R}_{\geq 0}$ by $f(S) = w(\overline{N}(S))$, where $\overline{N}(S) = \{v \in R \mid \delta(v) \subseteq S\}$. We initialize the density parameter as $\lambda_0 = f(S_0)/|S_0|$, and then minimize the function $\lambda_0|S| - f(S)$ over subsets $S \subseteq L$. Let $S_1$ be the minimizer of this objective, and set $\lambda_1 = f(S_1)/|S_1|$. This process is repeated iteratively. We will show how each minimization step can be performed using a single min-cut (or, equivalently, max-flow) computation. Algorithm 3 summarizes this iterative process.

---

**Algorithm 3** Density Improvement via Min-Cut

---

**Require:** Left vertex set $L$, right vertex set $R$, weight function $w : R \to \mathbb{R}_{\geq 0}$, neighborhood function $\delta : R \to 2^L$
1: Define $\overline{N}(S) \leftarrow \{v \in R \mid \delta(v) \subseteq S\}$
2: Define $f(S) \leftarrow \sum_{v \in \overline{N}(S)} w(v)$
3: Initialize $S_0 \leftarrow L, t \leftarrow 1$
4: Compute $\lambda_0 \leftarrow f(S_0)/|S_0|$
5: **repeat**
6:     Define $\Phi(S) \leftarrow \lambda_{t-1} \cdot |S| - f(S)$
7:     Find $S_t \subseteq L$ minimizing $\Phi(S)$ via a min-cut or max-flow computation
8:     Compute $\lambda_t \leftarrow f(S_t)/|S_t|$
9:     $t \leftarrow t + 1$
10: **until** $S_t = S_{t-1}$
11: **return** $S_t$

---

We now show how to minimize $\Phi(S) = \lambda|S| - f(S)$ for a fixed $\lambda$.

**Theorem 23.** *We can minimize $\Phi(S) = \lambda|S| - f(S)$ for a fixed $\lambda$ using a single minimum s-t cut.*

*Proof.* Construct a directed graph $G = (\{s, t\} \cup L \cup R, E)$ as follows:

- Add an edge from $s$ to each $u \in L$ with capacity $\lambda$.

- For each edge $(u, v)$ with $u \in L, v \in R$, if $v \in \delta(u)$, add an edge $u \to v$ with capacity $\infty$.

- Add an edge from each $v \in R$ to $t$ with capacity $w(v)$.

Consider an $s$-$t$ cut $(A, B)$, with $s \in A$, $t \in B$. Define the subset $S = L \cap B$, i.e., the left-hand vertices that are **cut off from** $s$ by the cut. We now compute the total capacity of the cut.

The cut capacity consists of:

1. Edges from $s$ to $S$, each with capacity $\lambda$, totaling $\lambda|S|$.

2. For each $v \in R$, the edge $v \to t$ is only cut if there exists any of its neighbors in $S$, i.e., $\delta(v) \cap S \neq \emptyset$, meaning $v \in \overline{N}(S)$ are not cut to $t$. The edge $v \to t$ contributes $w(R) - w(\overline{N}(S))$ to the cut.

Thus, the total cut capacity is:

$$\lambda|S| + w(R) - w(\overline{N}(S))$$

To minimize $\Phi(S) = \lambda|S| - f(S)$, we observe that this is equivalent (up to constants) to minimizing $\lambda|S| + w(R) - w(\overline{N}(S))$ since adding $w(R)$ from both sides doesn't change the argmin. Hence, the minimum $s$-$t$ cut yields a subset that minimizes $\Phi(S)$, as claimed. $\square$

# C   HNSN Experiments Results

**Problem Definition.** Recall the HNSN problem from the introduction. Since the function $f(S) = |N(S)|$ is submodular, HNSN is a special case of the weighted USSS problem. Ling *et al.* further reformulate the problem over the left vertex set $L$, defining the function $\overline{N} : 2^L \to \mathbb{R}$ by $\overline{N}(S) = \{v \in R \mid \delta(v) \subseteq S\}$ where $\delta(v)$ denotes the set of neighbors of $v$. Under this formulation, the problem becomes that of maximizing $w(\overline{N}(S))/|S|$ over non-empty subsets $S \subseteq L$. We adopt this viewpoint in our implementation.

**Algorithms.** We compare against the six baselines of Ling *et al.* [42], and introduce a new flow-based algorithm (detailed in Appendix B). In total, we evaluate **ten algorithms**: IP (SUPERGREEDY++), FW (Frank-Wolfe), FW-MNP (Fujishige-Wolfe Minimum Norm Point Algorithm), FLOW (our new flow-based method), CD (ContractDecompose, [42]), LP (Linear Programming, solved via Gurobi with academic license), GAR (Greedy Approximation, [42]), GR (Greedy, [42]), FGR (Fast Greedy, [42]), and GRR (GreedRatio, [5]). The first four are our implementations; the remaining six follow Ling *et al.*'s code [42].

**Approach.** For the three iterative algorithms—IP, FW, and FW-MNP—we run each for 100 iterations and plot the following: (1) the time taken per iteration, (2) the best density found up to that point. We report only the final runtime and the density achieved at termination for the remaining algorithms. In the runtime comparison, we include only those algorithms that (1) terminated within the time limit and (2) successfully found the optimal solution.

Our implementation of IP is custom and includes an optimized method for computing marginals. It maintains a heap over a set $S \subseteq L$ of vertices to peel from and a set $S' \subseteq R$ where $\delta(v) \subseteq S$ for all $v \in S'$. Once a vertex $\ell \in S$ is peeled, all its neighbors in $R$ are removed from the data structure $S'$. This allows each iteration to be implemented in $O(m \log n)$ time, where $m$ and $n$ are the number of edges and vertices in the bipartite graph, respectively.

**Datasets.** Table 1 summarizes the datasets used for the HNSN problem, following the benchmarks introduced by Ling *et al.* [42]. Each dataset is represented as a weighted bipartite graph with left vertices $L$, right vertices $R$, weights on the right vertices $w : R \to \mathbb{R}_{\geq 0}$, and edges $E$. The graphs are derived from diverse real-world domains, including recommendation systems (e.g., YooChoose, Kosarak), citation networks (ACM), social networks (NotreDame, IMDB, Digg), and financial transactions (e.g., E-commerce, Liquor). The datasets exhibit a wide range of sizes, with up to $\sim 10^6$ vertices and $\sim 2 \times 10^7$ edges.

**Resources**: In total, for all algorithms and all datasets, we request 4 CPUs per node and 35G of memory per experiment. All algorithms were given a 30-minute time limit per dataset, after which they were marked as Time Limit Exceeded and given blank values in the plots below.

**Discussion of HNSN Results.** See Figure 2 for all plots. For each dataset, the top plot shows the final density found by each algorithm (the higher, the better), and the bottom plot shows the time each algorithm takes on a **log-scale** (the lower, the better). We exclude algorithms with a time limit (more than 30 minutes) or if they give a suboptimal answer. Across all datasets, the Frank-Wolfe (FW) algorithm and Fujishige-Wolfe Minimum Norm Point (FW-MNP) algorithm consistently emerged as the fastest to converge. In nearly every case, FW-MNP outperformed FW, achieving the best solution quality in the shortest time, with only a single dataset where FW was slightly faster. Relative to the six baselines from Ling *et al.* [42], these methods achieved speedups ranging from $5\times$ to $595\times$.

Our new flow-based algorithm and SUPERGREEDY++ (IP) typically followed, ranking third and fourth in performance across datasets. Figure 1a shows a particularly striking example, where on the YooChoose dataset, FW-MNP achieved a $595\times$ speedup over the best previously published baseline. In this instance, all algorithms converged to the correct final density, except for GRR, which terminated at a suboptimal solution and was therefore excluded from the runtime comparison.

In particular, the algorithms GRR, GR, GAR, and LP frequently encountered difficulties, either due to slow performance (time limit exceeding) or convergence to incorrect solutions. These results underscore a recurring insight consistently observed across our experiments: universal solvers such as FW-MNP and FW, originally developed for broader optimization tasks, can significantly outperform specialized heuristics when applied to specific problems like HNSN, provided they are used within the proper conceptual framework that we have outlined.

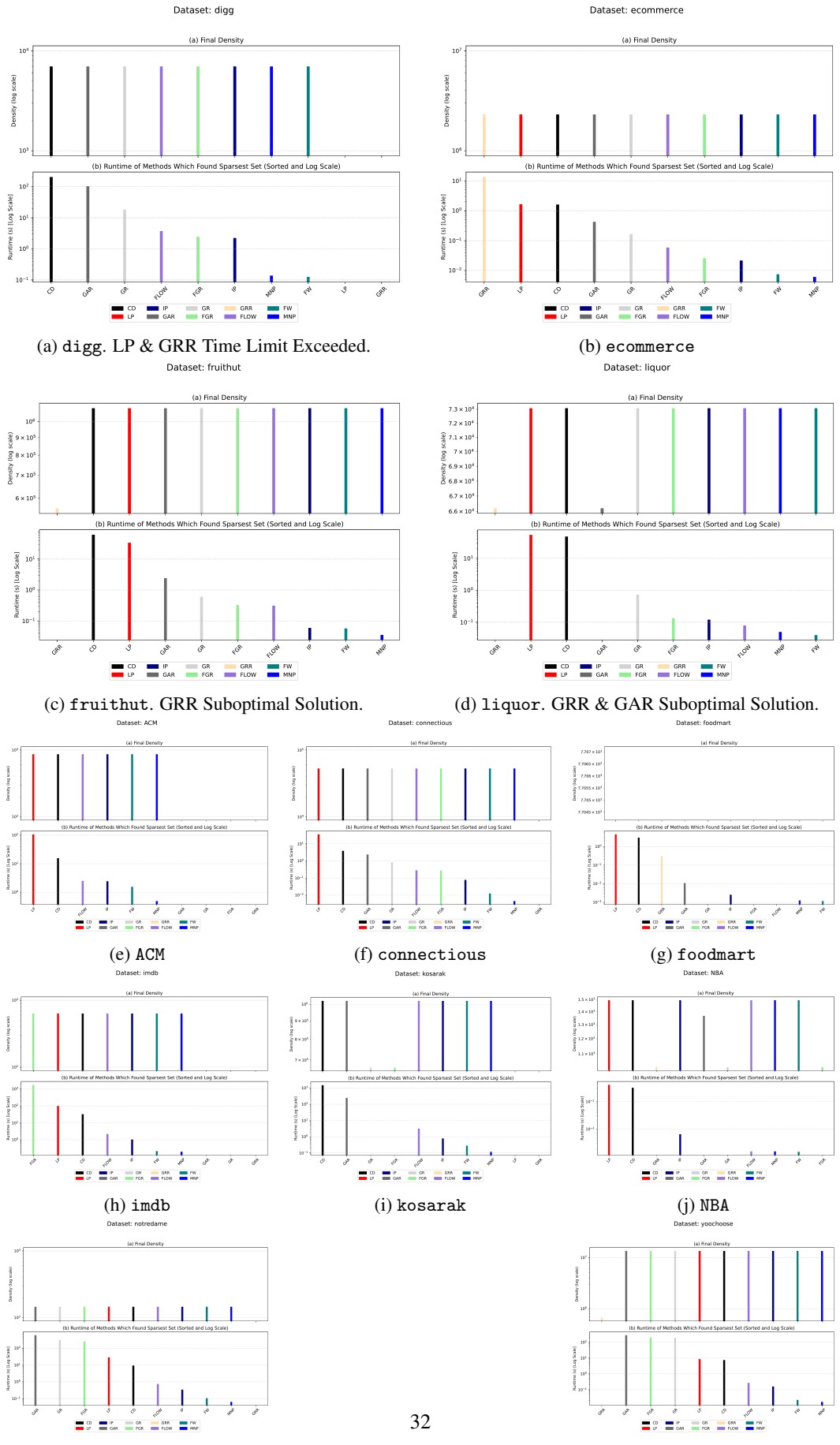

(a) `digg`. LP & GRR Time Limit Exceeded.

(b) `ecommerce`

(c) `fruithut`. GRR Suboptimal Solution.

(d) `liquor`. GRR & GAR Suboptimal Solution.

(e) `ACM`

(f) `connectious`

(g) `foodmart`

(h) `imdb`

(i) `kosarak`

(j) `NBA`

(k) `notredame`

(l) `yoochoose`

Figure 2: HNSN results across 12 datasets.

## D  Min $s$-$t$ Cut Experiments

**Problem Definition.**  We next evaluate our algorithms on the classical *Minimum s-t Cut* problem, a fundamental task in combinatorial optimization with widespread applications. Specifically, the Minimum $s$-$t$ Cut problem can be framed as minimizing the normalized submodular function $g(S) = |\delta(S \cup \{s\})| - |\delta(\{s\})|$ over subsets $S \subseteq V \setminus \{s, t\}$, ensuring that any feasible solution corresponds to a valid $s$-$t$ cut. Within this formulation, the Minimum $s$-$t$ Cut problem becomes an instance of *Submodular Function Minimization (SFM)*, and thus naturally fits within the unified framework established in this paper. This connection allows us to apply universal solvers, such as SUPERGREEDY++, Frank-Wolfe, and Wolfe's Minimum Norm Point algorithm, to this classical combinatorial problem.

**Algorithms.**  We evaluate three iterative algorithms—Frank-Wolfe (FRANKWOLFE), Fujishige-Wolfe Minimum Norm Point (FW-MNP), and SUPERGREEDY++ (SUPERGREEDY)—on the Minimum $s$-$t$ Cut problem. For each instance, we also compute the exact minimum cut value using standard combinatorial flow-based methods (*Edmonds Karp algorithm*), serving as ground truth. This allows us to assess both the solution quality and runtime efficiency of the optimization-based approaches relative to the exact combinatorial solution. We run SUPERGREEDY++ for 1000 iterations on each dataset, 10,000 iterations for FRANKWOLFE, and 500 for FW-MNP.

**Marginals.**  The iterative algorithms leverage the marginal $g(v|S - v) = g(S) - g(S - v)$; the difference in the cut value when moving $v$ from the source partition to the sink partition. It follows that $g(v|S - v)$ is realized by $\deg^+(v) - \deg^-(v)$ where $\deg^-(v)$ is the (weighted) degree of $v$ to all vertices $u \in S \cup \{s\}$ and $\deg^+(v)$ is the (weighted) degree of $v$ to all vertices $w \in V \setminus (S \cup \{s\})$.

**Datasets.**  Our evaluation spans two groups of datasets. First, we consider four classical benchmark instances from the first DIMACS Implementation Challenge on minimum cut [21], known for their small size but challenging cut structures. Second, to assess scalability and robustness, we include a large-scale dataset family, specifically the BVZ-SAWTOOTH instances [38], which consist of 20 distinct minimum $s$-$t$ cut problems. Each instance contains approximately 500,000 vertices and 800,000 edges, providing a test of algorithmic scalability.

**Resources**:  In total, for all algorithms and all datasets, we request 4 cpus per node, and 40G of memory per experiment. All algorithms were given a 25-minute time limit per dataset (all algorithms finished within this time, and there was no time limit exceeded). We mark with a vertical dotted line the time for the flow algorithm to find the correct minimum cut.

**Discussion of Minimum $s$-$t$ Cut Results.**  Full results are provided in Figure 3 and 4. Across all datasets, SUPERGREEDY++ was consistently the fastest to approximate the minimum cut, often by large margins over exact flow-based methods, FW-MNP, and FW. In 16 of 24 instances, it found the exact min-cut rapidly. In the remaining cases, its solution was within $1.000023\times$ the optimum after at most 500 iterations, always converging before the exact flow solver. A notable example is the BVZ-SAWTOOTH14 instance, where SUPERGREEDY++ converges orders of magnitude faster (Figure 1c). This strong performance parallels its success in dense subgraph problems, where it efficiently finds high-quality solutions but can slow down near the optimum. Combining SUPERGREEDY++ with flow-based refinement remains an interesting direction for future work. *Interestingly, while* FW-MNP *and* FW *excel on* HNSN, *they both underperform here, underscoring how problem structure affects solver efficiency*.

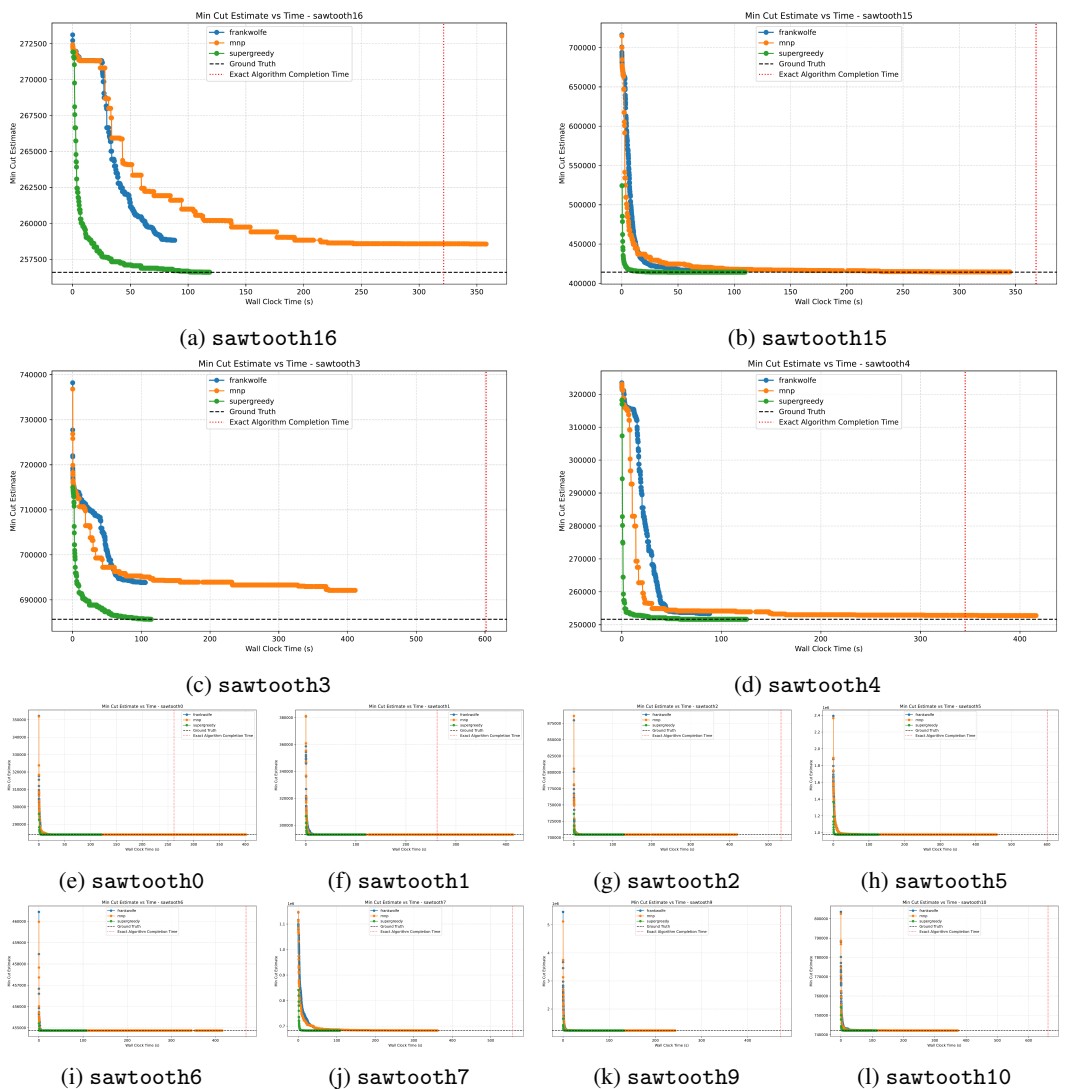

Figure 3: Min-Cut vs Wall Clock Time (Part 1).

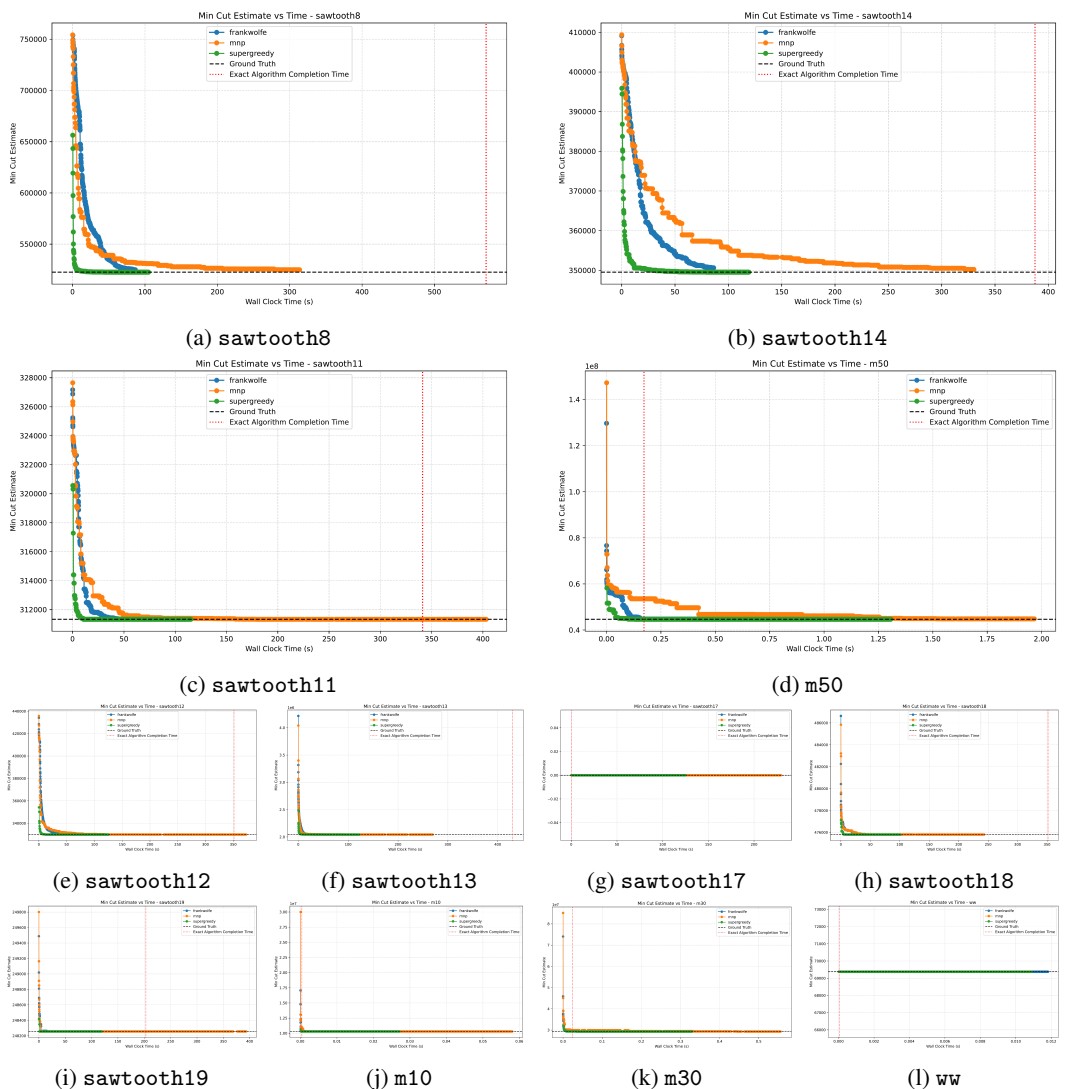

(a) `sawtooth8`

(b) `sawtooth14`

(c) `sawtooth11`

(d) `m50`

(e) `sawtooth12`

(f) `sawtooth13`

(g) `sawtooth17`

(h) `sawtooth18`

(i) `sawtooth19`

(j) `m10`

(k) `m30`

(l) `ww`

Figure 4: Min-Cut vs Wall Clock Time (Part 2).

# E  Densest Subgraph Problem Experiments

**Problem Definition.** Given a graph $G = (V, E)$, find a vertex set $\emptyset \neq S \subseteq V$ maximizing $|E(S)|/|S|$.

**Algorithms.** We evaluate *seven* algorithms—Frank-Wolfe (FRANKWOLFE, [19]), Fujishige-Wolfe Minimum Norm Point (FW-MNP), and SUPERGREEDY++ (SUPERGREEDY, [9]), FISTA [31], RCDM [47], the incremental algorithm [33], and a new push-relabel flow-based algorithm based on the density improvement mechanism, but using push-relabel flow.

Table 3: Summary of Densest Subgraph Datasets.

| Dataset | # Nodes | # Edges |
|---|---|---|
| close-cliques | 3,230 | 95,400 |
| com-amazon | 334,863 | 925,872 |
| com-dblp | 317,080 | 1,049,866 |
| com-orkut | 3,072,441 | 117,185,083 |
| disjoint_union_Ka | 35,900 | 16,158,800 |
| roadNet-PA | 1,088,092 | 3,083,796 |
| roadNet-CA | 1,965,206 | 5,533,214 |

**Datasets.** Table 3 summarizes the 7 datasets we used for the densest subgraph experiments, two of which are synthetic from [31].

**Resources**: In total, for all algorithms and all datasets, we request 4 CPUs per node and 40G of memory per experiment. The only exception is for the dataset com_orkut, where we requested 100G of memory. All algorithms were given a 30-minute time limit per dataset; otherwise, they were marked as having exceeded the time limit.

**Discussion of Densest Subgraph Results.** Figure 5 presents the evolution of subgraph density over time for all datasets in the classic densest subgraph problem. Each plot includes the full runtime and a zoomed-in view of the first 20% execution time to highlight early algorithm behavior.

Some literature on densest subgraph algorithms has lacked consistent evaluations, with many works often reaching conflicting conclusions about which algorithms are the most efficient. We aim to contribute a more nuanced and systematic perspective on performance evaluation. We hope future work adopts similar care when assessing and comparing algorithms for the densest subgraph problem.

**Memory Comparison:** Flow-based algorithms consistently delivered strong performance, particularly when paired with the iterative density-improvement framework of Veldt et al. [34] and Hochbaum [33]. However, a significant practical drawback is their memory usage: in our experiments, flow-based methods often consumed 2–3× more memory than other approaches. This overhead stems from the substantial bookkeeping in constructing and maintaining the flow network.

Moreover, flow-based methods incurred noticeable delays during the initial stages due to the cost of constructing flow networks on large graphs. While Hochbaum employs the Pseudoflow algorithm, we observed comparable performance using standard Push-Relabel solvers within the same density-improvement framework, reaffirming that the primary source of speedup lies in the iterative framework itself rather than the specific max-flow implementation.

That said, flow-based methods are not universally applicable. In problems such as the generalized $p$-mean Densest Subgraph, no known linear flow formulations minimize objectives like $\lambda|S| - f(S)$ for arbitrary supermodular functions. In such cases, algorithms like SUPERGREEDY++, Frank-Wolfe, and the Fujishige-Wolfe Minimum Norm Point algorithm (FW-MNP) remain the only viable choices, as they operate purely via oracle access and require no specialized structure.

**Convergence to a $(1 - \varepsilon)$ approximation:** For fast convergence to near-optimal solutions, convex programming-based methods were clear standouts. Across nearly all datasets, RCDM consistently reached $(1 - \varepsilon)$-approximate dense subgraphs in just a few iterations. It was closely followed by FISTA and SUPERGREEDY++, which also showed rapid convergence and high-quality intermediate solutions.

However, it is essential to note that RCDM and FISTA rely on a projection oracle from [31], which may not be available for generalized objectives (e.g., $p$-mean Densest Subgraph). In contrast, SU-

PERGREEDY++ maintains its effectiveness without requiring such oracles, making it more broadly applicable.

On the other hand, Frank-Wolfe and Fujishige-Wolfe's FW-MNP Algorithm often required thousands of iterations to achieve a reasonable approximation and were generally outperformed by other methods. While their theoretical underpinnings are strong, their practical convergence to high-quality solutions is often prohibitively slow.

**Convergence to the optimal solution:** When the goal is **exact optimality**, flow-based methods paired with the density-improvement framework outperformed all other algorithms by a wide margin. Even though convex programming methods like RCDM, FISTA, and SUPERGREEDY++ rapidly approached near-optimal values, they often stalled and required hundreds of iterations to fully converge to the exact solution *on some datasets*.

In contrast, the flow-based approaches often converged to the optimal solution orders of magnitude faster, especially on mid-sized and large graphs. Notably, this speed is not a consequence of the specific flow algorithm used but instead of the density-improvement strategy, which efficiently reduces the problem to a sequence of min-cut instances.

Thus, flow-based methods remain the most effective choice for applications where exact solutions are necessary, provided memory usage is manageable and a suitable flow reduction exists.

**General Takeaways:** Our experiments highlight a few key insights:

1. Flow-based algorithms remain among the most potent tools for the exact densest subgraph discovery, especially when integrated with iterative frameworks, but they require significant memory and do not apply to generalized objectives.

2. Convex optimization methods like RCDM and FISTA converge rapidly to high-quality approximate solutions, making them ideal for problems with projection oracles and when approximate results suffice.

3. SUPERGREEDY++ offers a strong balance of speed, quality, and generality, consistently performing well across both real and synthetic datasets and requiring no specialized problem structure.

4. Finally, algorithm selection should be guided by the problem setting: whether exactness is required, whether a flow reduction exists and whether projection oracles are available.

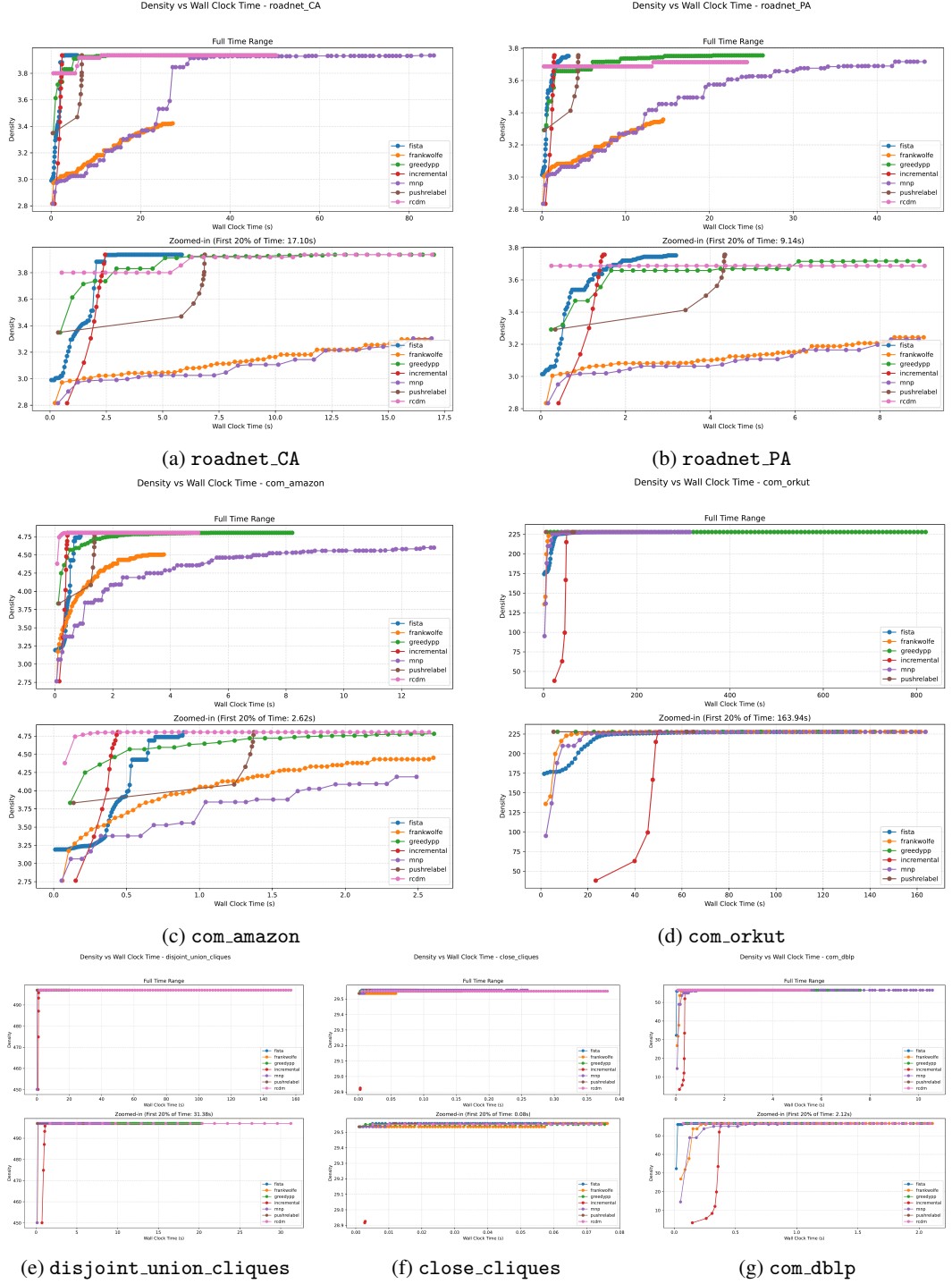

Figure 5: DSG density over time

# F  Generalized $p$-mean Densest Subgraph Experiments

**Problem Definition.** Given a graph $G = (V, E)$, find a vertex set $\emptyset \neq S \subseteq V$ maximizing the generalized density $\left( \frac{1}{|S|} \sum_{v \in S} \deg_S^p(v) \right)^{1/p}$.

**Algorithms.** We evaluate three algorithms—Frank-Wolfe (FRANKWOLFE), Fujishige-Wolfe Minimum Norm Point (FW-MNP), and SUPERGREEDY++ (SUPERGREEDY). All algorithms are implemented in C++.

**Marginals.** We adopt the approach from [58], noting that maximizing the objective is equivalent to maximizing $f(S)/|S|$, where $f(S) = \sum_{v \in S} \deg_S^p(v)$ which is supermodular. Peeling of any vertex affects the denominator equivalently; hence, we act greedily with respect to the numerator. The marginal cost of peeling a vertex $v$ from $S$ is then given by

$$f(v|S - v) = f(S) - f(S - v) = \deg_S^p(v) + \sum_{u \in \delta_S(v)} [\deg_S^p(u) - (\deg_S(u) - 1)^p]$$

where $\delta_S(v)$ denotes $v$'s neighbours in $S$.

Table 4: Summary of Generalized $p$-mean Densest Subgraph Datasets.

| Dataset | # Nodes | # Edges |
|---|---|---|
| close-cliques | 3,230 | 95,400 |
| com-amazon | 334,863 | 925,872 |
| com-dblp | 317,080 | 1,049,866 |
| roadNet-PA | 1,088,092 | 3,083,796 |
| roadNet-CA | 1,965,206 | 5,533,214 |

**Datasets.** Table 4 summarizes the 5 datasets we used for the generalized $p$-mean Densest Subgraph experiments. Note that these are a subset of the datasets we used for DSG. We consider four values for $p \in \{1.1, 1.25, 1.5, 1.75\}$.

**Resources.** We request 4 CPUs per node and 40G of memory per experiment across all algorithms and datasets. Each trial is given a 30-minute time limit; none exceeded this limit.

**Discussion of generalized $p$-mean Densest Subgraph Results.** Results for all datasets are presented in Figures 6 to 10. We plot the best density found against wall-clock time, and, as in previous sections, each plot includes both the full runtime and a zoomed-in view covering the first 20% of execution time. Overall, we find that SUPERGREEDY++ generally achieves the highest density within the allotted time and tends to converge more quickly than the other algorithms.

On the close_cliques dataset, however, FW and FW-MNP demonstrate stronger early-stage performance, reaching the maximum density faster than SUPERGREEDY++, though the latter eventually converges to the same value. By contrast, SUPERGREEDY++ often converges within the first few iterations on real-world instances, while the other two algorithms trail behind. Relative performance between FW and FW-MNP varies, with each occasionally outperforming the other.

We also observe that increasing the value of $p$ in the generalized objective improves the performance of both FW and FW-MNP, whereas SUPERGREEDY++ remains largely unaffected. In fact, for all values of $p$ considered, both FW and FW-MNP show improved performance compared to the classical Densest Subgraph case ($p = 1$).

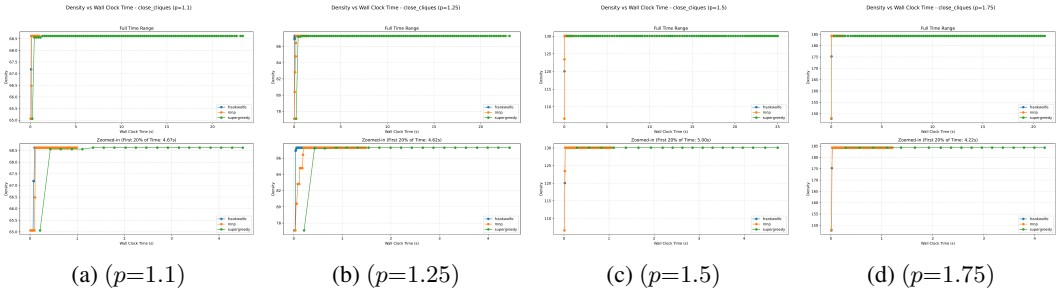

(a) ($p$=1.1)    (b) ($p$=1.25)    (c) ($p$=1.5)    (d) ($p$=1.75)

Figure 6: DSS density over time - close_cliques

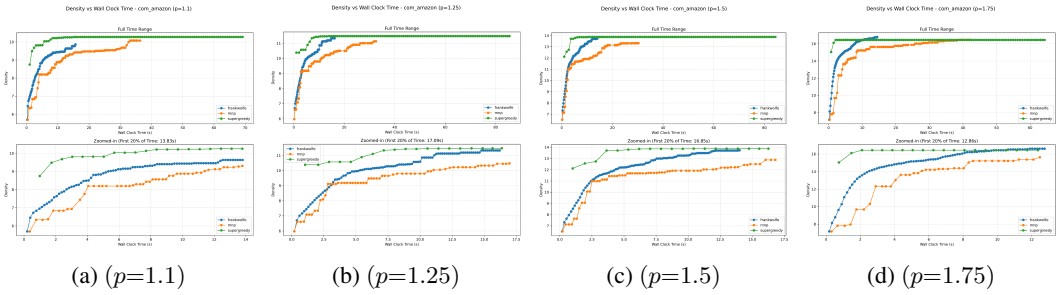

(a) ($p$=1.1)      (b) ($p$=1.25)      (c) ($p$=1.5)      (d) ($p$=1.75)

Figure 7: DSS density over time - `com_amazon`

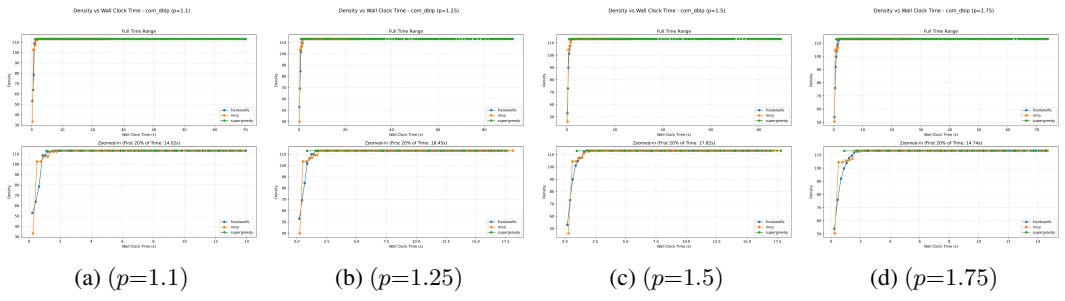

(a) ($p$=1.1)      (b) ($p$=1.25)      (c) ($p$=1.5)      (d) ($p$=1.75)

Figure 8: DSS density over time - `com_dblp`

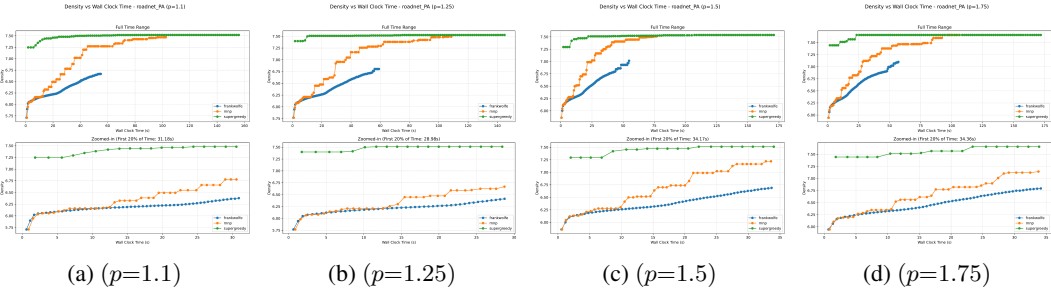

(a) ($p$=1.1)      (b) ($p$=1.25)      (c) ($p$=1.5)      (d) ($p$=1.75)

Figure 9: DSS density over time - `roadnet_pa`

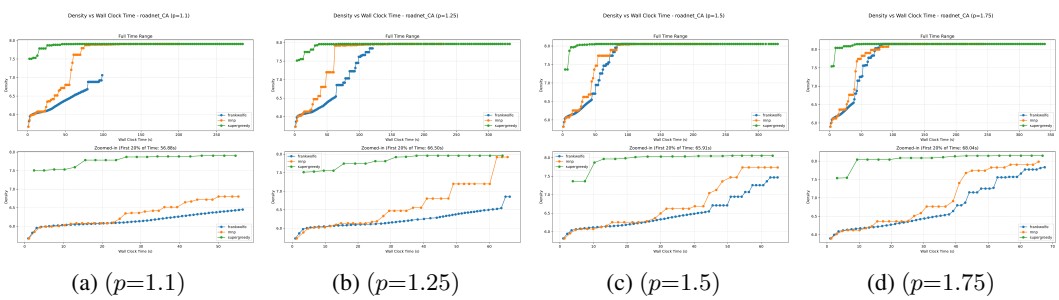

(a) ($p$=1.1)      (b) ($p$=1.25)      (c) ($p$=1.5)      (d) ($p$=1.75)

Figure 10: DSS density over time - `roadnet_ca`

# G  Anchored Densest Subgraph

**Problem Definition.** Given a graph $G = (V, E)$ and a set $R \subseteq V$, find a vertex set $\emptyset \neq S \subseteq V$ maximizing $\left(2|E(S)| - \sum_{v \in S \cap \overline{R}} \deg_G(v)\right)/|S|$.

**Algorithms.** We evaluate four algorithms—Frank-Wolfe (FRANKWOLFE), Fujishige-Wolfe Minimum Norm Point (FW-MNP), and SUPERGREEDY++ (SUPERGREEDY), and the flow-based algorithm by Huang *et al.* [34]. The authors' code was implemented in Julia, so we re-implement all algorithms in C++.

**Marginals.** Similar to the generalized $p$-mean DSG, we recognize that maximizing the objective can be achieved by acting greedy with respect to the numerator where $|E(S)|$ is supermodular, and the degree sum is modular as it's with respect to $G$. Hence, the objective is supermodular. The marginal cost of peeling $v$ is then

$$f(v|S - v) = 2 \deg_S(v) - \mathbb{I}_{v \notin R}[\deg_G(v)]$$

**Datasets.** We reused the base graphs from our generalized $p$-mean Densest Subgraph Experiments, summarized in Table 4. These are the initial graphs $G$ given, for which anchor sets $R$ are created via random walks.

**Generating Random $R$.** As in [34], we generate $R$ with the following process: first, we sample 10 seed nodes from the set of vertices uniformly at random. Next, using their 2-neighborhood, we sample more nodes using a random walk until the total number of nodes in $R$ are 201, and mark that set as $R$.

**Resources**: In total, for all algorithms and all datasets, we request 4 CPUs per node and 40G of memory per experiment. All algorithms were given a 10-minute time limit per dataset (all algorithms finished within this time, and no time limit was exceeded except on one dataset for the flow-based algorithm).

**Discussion of Anchored Densest Subgraph Results.** Figure 11 summarizes the results for anchored densest subgraph. The top plot shows results over the entire runtime, while the bottom plot zooms in on the first 20% to highlight early iterations. For each dataset, we plot the density of the best-anchored subgraph found against wall-clock time.

Across all experiments, SUPERGREEDY++ consistently delivered the best performance, achieving the highest densities in the shortest time. It was followed by FW and FW-MNP, which ranked second and third overall, though their relative performance varied across datasets—FW often had the edge over FW-MNP, but not uniformly. The flow-based method, FLOW, performed significantly worse on most datasets (with the notable exception of com_amazon). This underperformance may be attributed to high variance in edge capacities within the flow network, which can cause the algorithm to stall when attempting to push flow efficiently. In future work, it is worth studying the performance of flow algorithms in the weighted case, where such capacity imbalances are more pronounced or may offer new opportunities for optimization.

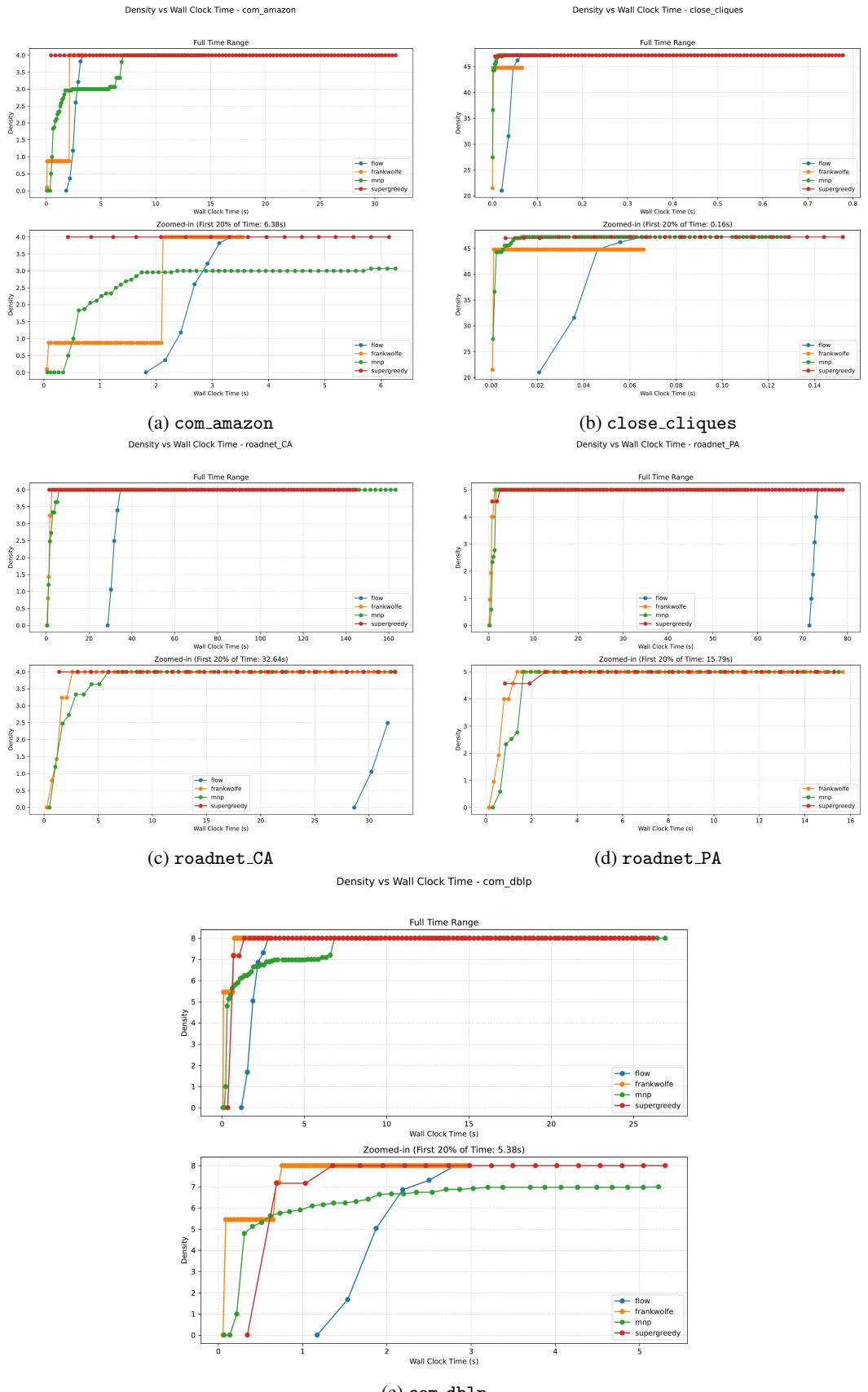

Figure 11: Anchored density over time

# H Contrapolymatroid Membership Experiments

**Problem Definition (CM).** Given a supermodular function $f : 2^V \to \mathbb{R}$ and arbitrary vector $y \in \mathbb{R}^{|V|}$, determine if $y \in B(f)$.

**Approach.** The problem can be reduced to an SFM instance by recognizing that $-f(S)$ is submodular, hence $x(S) - f(S)$ is also submodular. We minimize $y(S) - f(S)$ to answer the membership question. It is more convenient to interpret this as maximization of $f(S) - y(S)$, whose value we plot; $y$ is then a NO instance *iff* $f(S) - y(S) > 0$.

For our experiments, we consider undirected graphs $G = (V, E)$ with $f(S) = |E(S)|$, $S \subseteq V$.

**Generating Query Vectors.** Harb et al. [31] (Theorem 4.4) showed that a vector $b \in B(f)$ if and only if there exists a vector $x \in \mathbb{R}^{2|E|}$ such that the pair $(x, b)$ satisfies the dual of Charikar's densest subgraph LP [16]. Such a vector $b$ corresponds to a YES instance of the Contrapolymatroid Membership (CM) problem. Small perturbations to $b$ can then be used to generate arbitrarily hard NO instances.

Given a graph $G = (V, E)$, we construct a feasible vector $b \in B(f)$ by orienting edges. We begin by computing the exact densest subgraph using PUSHRELABEL. For each vertex $v \in S^*$—the vertex set of the densest subgraph—we set $b_v = \lambda^*$, where $\lambda^*$ is the maximum density. Harb et al. [31] (Theorem 4.1) show that such an assignment is always realizable via some orientation of the edges in $S^*$. For edges not entirely contained in $S^*$, we assign a value of 0.5 to each endpoint if both lie outside $S^*$ and a value of 1 to the endpoint outside $S^*$ otherwise.

To generate non-trivial NO instances (i.e., where $x(V) = f(V)$), we perturb the vector as follows: select two vertices $u \in S^*$ and $w \notin S^*$, then set $b_u \leftarrow b_u - \varepsilon$ and $b_w \leftarrow b_w + \varepsilon$, where $\varepsilon$ is the perturbation magnitude. This results in

$$x(S^*) = |S^*|\lambda^* - \varepsilon = |E(S^*)| - \varepsilon < |E(S^*)| = f(S^*),$$

violating the contrapolymatroid inequality for at least the set $S^*$.

In our experiments, we reuse the generalized $p$-mean DSG setup with $p = 1$, using the supermodular objective $f(S) = \sum_{v \in S} \deg_S(v)$. Note that $|E(S)| - y(S) > 0$ if and only if $\sum_{v \in S} \deg_S(v) - 2y(S) > 0$. We maximize and report the latter expression, which is always at least $2\varepsilon$, with the lower bound attained when $S = S^*$.

**Algorithms.** We evaluate three algorithms—Frank-Wolfe (FW), Fujishige-Wolfe Minimum Norm Point (FW-MNP), and SUPERGREEDY++ (SUPERGREEDY). All algorithms are implemented in C++.

**Datasets.** We reuse the datasets summarized in Table 4. We generate four increasingly difficult NO instances for each dataset with perturbation magnitudes $\varepsilon \in \{12, 6, 1, 1\mathrm{e}{-1}\}$.

**Resources.** In total, for all algorithms and all datasets, we request 4 CPUs per node and 40G of memory per experiment. All algorithms were given a 30-minute time limit per dataset (all algorithms finished within this time, and no time limit was exceeded).

**Discussion of Contrapolymatroid Membership Results.** Figures 12-16 summarize the results. An algorithm detects a NO instance when its value *climbs* from 0 to a non-zero value. The CM results continue the observed trend: SUPERGREEDY++ performs strongly in general but is outperformed by FW and FW-MNP on CLOSE_CLIQUES, where SUPERGREEDY++ fails to identify NO instances until the perturbation is sufficiently large ($\varepsilon \geq 6$), and even then, does so more slowly. For all other instances, SUPERGREEDY++ is usually the first—and frequently the only—algorithm to identify NO instances, often outpacing FW and FW-MNP by a wide margin. SUPERGREEDY++ also tends to find larger values, though this is irrelevant for CM. As expected, performance improves with larger $\varepsilon$.

This strong empirical performance of SUPERGREEDY++ is particularly surprising given that Contrapolymatroid Membership was the original motivation for studying submodular function minimization, and FW-MNP has historically been regarded as the algorithm of choice—making it especially noteworthy that a combinatorial method like SUPERGREEDY++ outperforms it so consistently across all but one dataset.

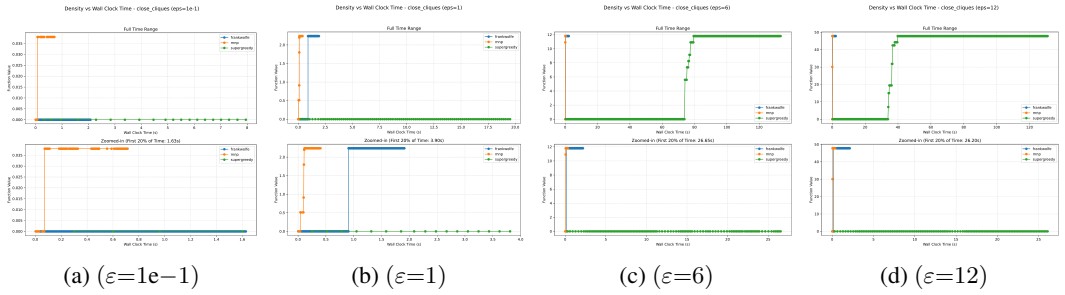

(a) ($\varepsilon$=1e−1)  (b) ($\varepsilon$=1)  (c) ($\varepsilon$=6)  (d) ($\varepsilon$=12)

Figure 12: Contrapolymatroid Membership Function Value over time - `close_cliques`

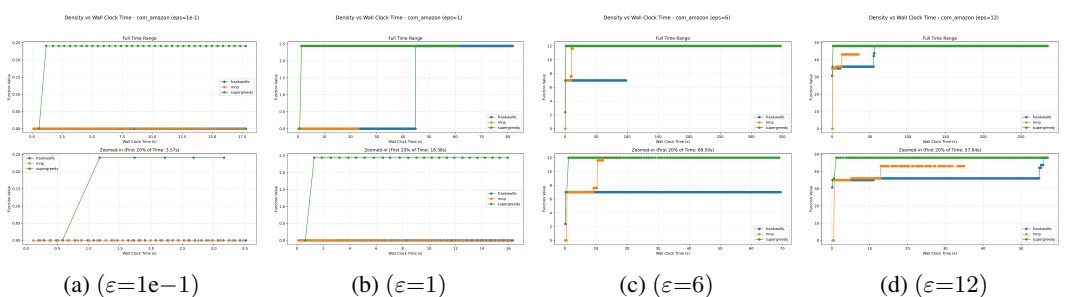

(a) ($\varepsilon$=1e−1)  (b) ($\varepsilon$=1)  (c) ($\varepsilon$=6)  (d) ($\varepsilon$=12)

Figure 13: Contrapolymatroid Membership Function Value over time - `com_amazon`

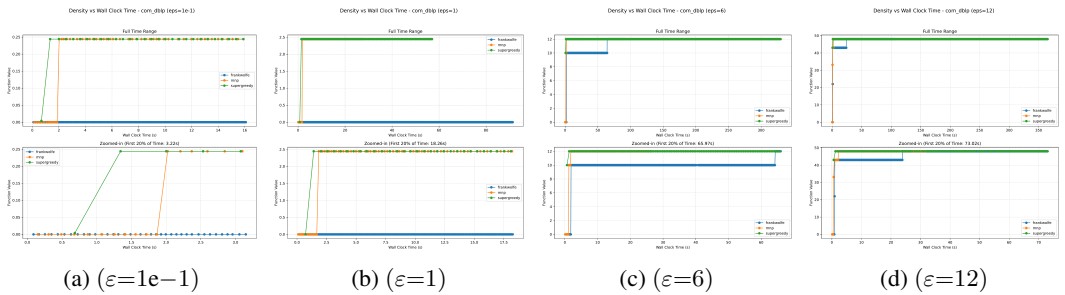

(a) ($\varepsilon$=1e−1)  (b) ($\varepsilon$=1)  (c) ($\varepsilon$=6)  (d) ($\varepsilon$=12)

Figure 14: Contrapolymatroid Membership Function Value over time - `com_dblp`

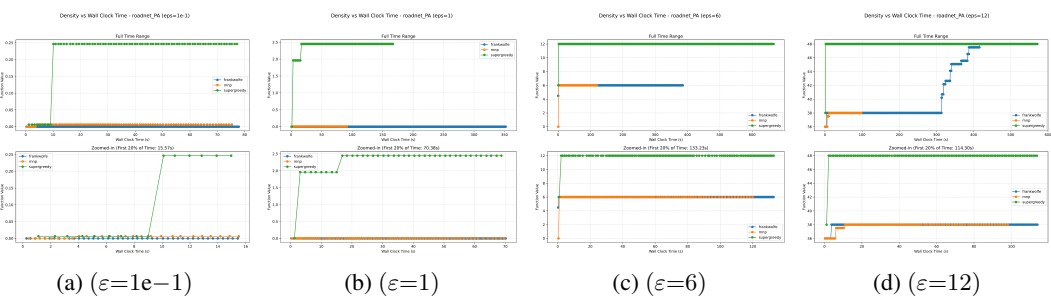

(a) ($\varepsilon$=1e−1)  (b) ($\varepsilon$=1)  (c) ($\varepsilon$=6)  (d) ($\varepsilon$=12)

Figure 15: Contrapolymatroid Membership Function Value over time - `roadnet_PA`

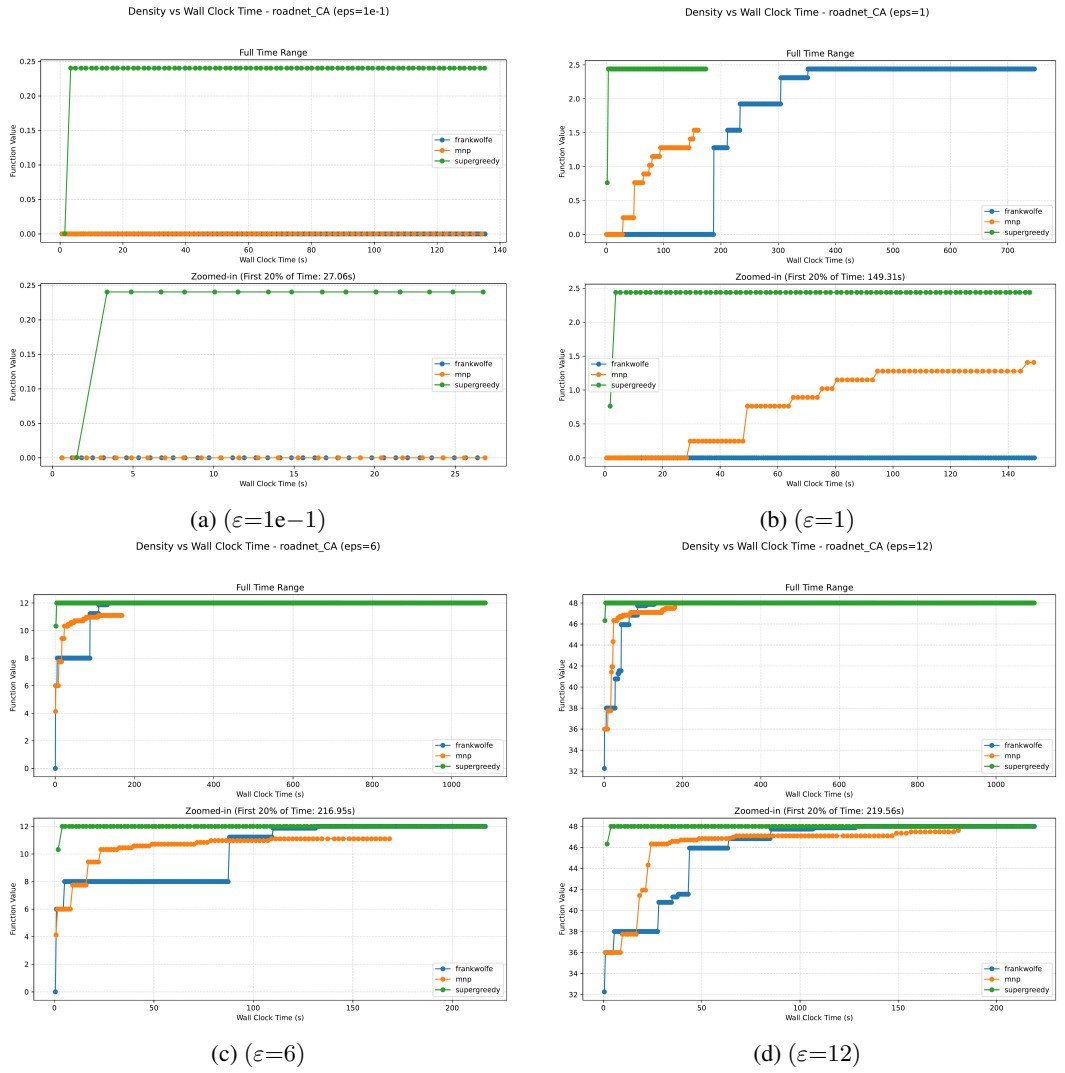

(a) $(\varepsilon{=}1e{-}1)$          (b) $(\varepsilon{=}1)$

(c) $(\varepsilon{=}6)$          (d) $(\varepsilon{=}12)$

Figure 16: Contrapolymatroid Membership Function Value over time - `roadnet_CA`

# I Minimum Norm Point Problem Experiments

**Problem Definition.** Let $f : 2^V \to \mathbb{R}$ be a normalized (super)submodular function, and let $B(f)$ denote its base polytope. Our objective is to find the point in $B(f)$ with the minimum Euclidean norm, i.e., $\arg\min_{x \in B(f)} \|x\|_2^2$.

**Algorithms.** We evaluate three algorithms: Frank-Wolfe (FRANKWOLFE), Fujishige-Wolfe Minimum Norm Point (FW-MNP), and SUPERGREEDY++ (SUPERGREEDY). All algorithms are implemented in C++.

**Datasets.** We reuse all instances from prior experiments but focus on the load vector norm rather than the original objective.

**Resources.** Please refer to the previously cited resources for further information on each problem and its resources.

**Discussion of Minimum Norm Point Results.** Algorithm performance varies significantly across problems and instances. Overall, SUPERGREEDY++ performs best on DSG, consistently achieving the lowest norm point; the other algorithms behave similarly and sub-optimally. For Anchored DSG, Generalized $p$-mean DSG, and HNSN, FW and FW-MNP typically outperform SUPERGREEDY++. However, all methods generally converge to similar norm values—except in generalized $p$-mean DSG, where SUPERGREEDY++ often settles at suboptimal norms despite excelling in the classical DSG case ($p = 1$). A comparable pattern appears in the `close_cliques` instance for Contrapolymatroid Membership, while in other CM instances, SUPERGREEDY++ converges faster and achieves lower norms. Lastly, for Min $s$-$t$ Cut, SUPERGREEDY++ and FW-MNP perform comparably, with SUPERGREEDY++ slightly outperforming, and both approaches consistently exceeding FW.

## I.1 DSG

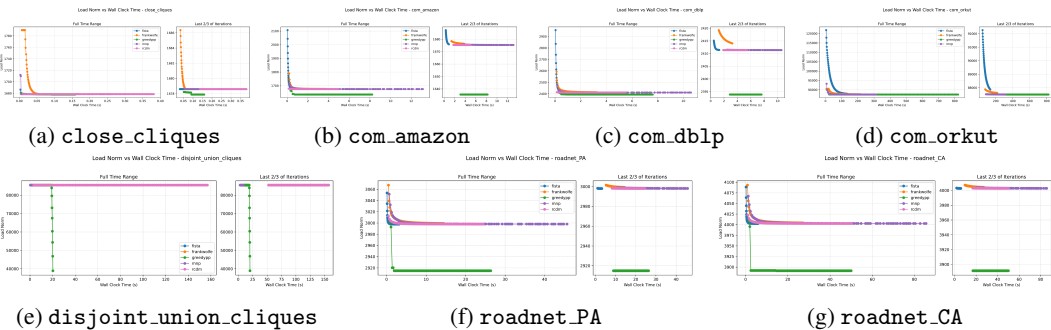

(a) `close_cliques`   (b) `com_amazon`   (c) `com_dblp`   (d) `com_orkut`

(e) `disjoint_union_cliques`   (f) `roadnet_PA`   (g) `roadnet_CA`

Figure 17: DSG load norm over time

## I.2 Anchored DSG

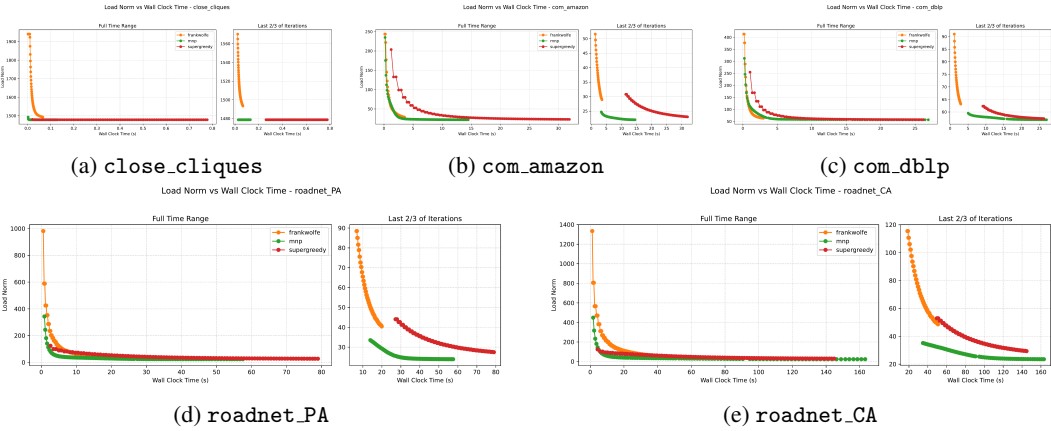

(a) `close_cliques`   (b) `com_amazon`   (c) `com_dblp`

(d) `roadnet_PA`   (e) `roadnet_CA`

Figure 18: Anchored DSG load norm over time

## I.3 HNSN

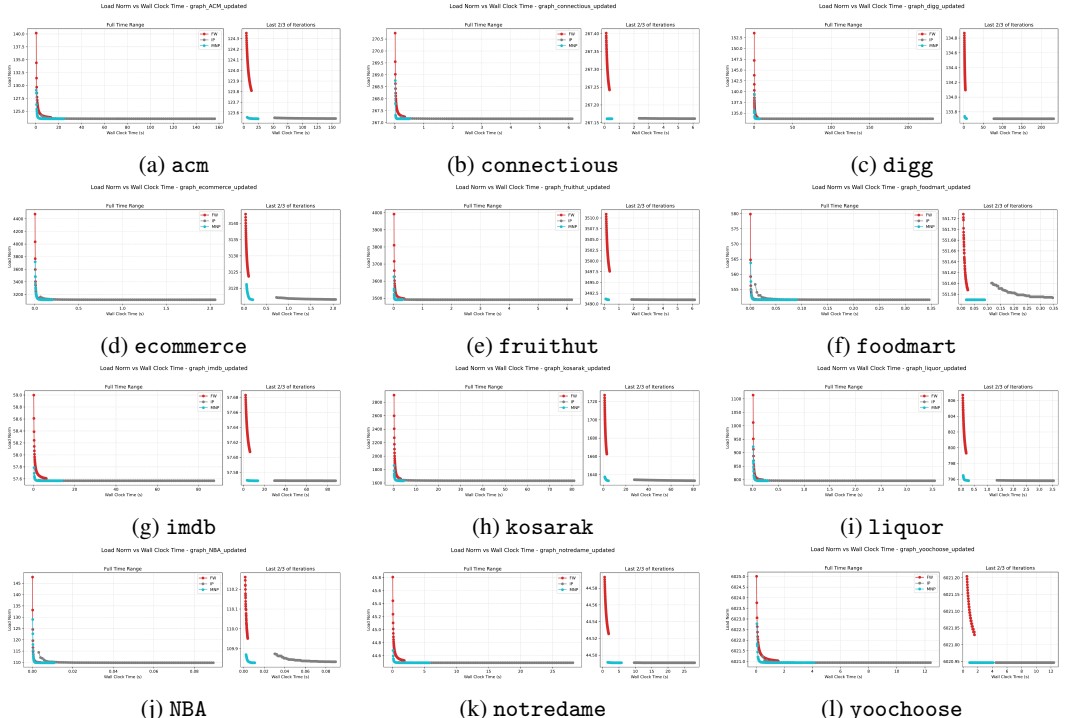

(a) `acm`

(b) `connectious`

(c) `digg`

(d) `ecommerce`

(e) `fruithut`

(f) `foodmart`

(g) `imdb`

(h) `kosarak`

(i) `liquor`

(j) `NBA`

(k) `notredame`

(l) `yoochoose`

Figure 19: HNSN load norm over time

## I.4 Min $s$-$t$ Cut

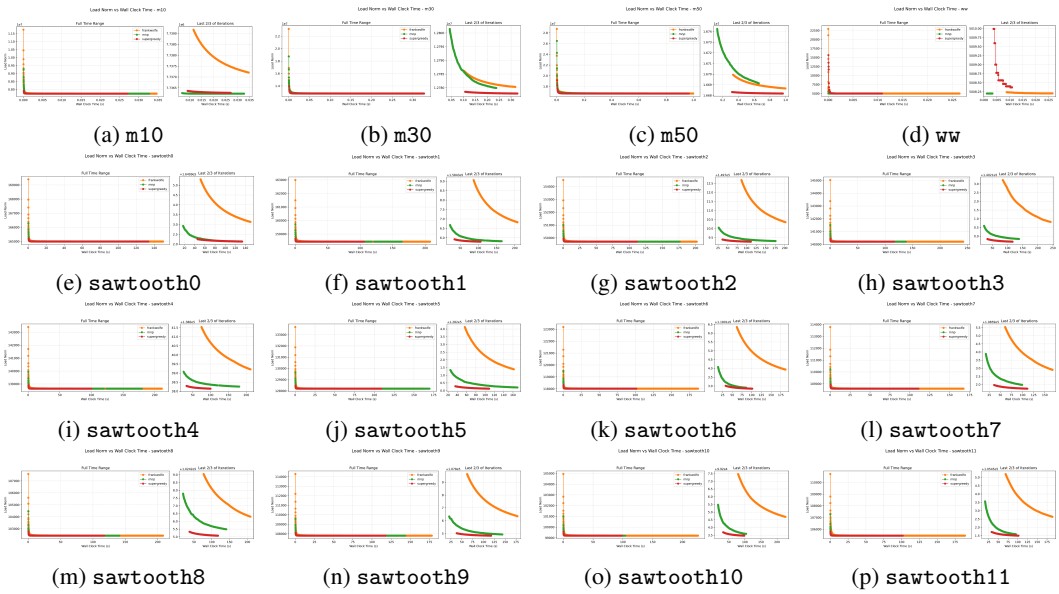

(a) `m10`

(b) `m30`

(c) `m50`

(d) `ww`

(e) `sawtooth0`

(f) `sawtooth1`

(g) `sawtooth2`

(h) `sawtooth3`

(i) `sawtooth4`

(j) `sawtooth5`

(k) `sawtooth6`

(l) `sawtooth7`

(m) `sawtooth8`

(n) `sawtooth9`

(o) `sawtooth10`

(p) `sawtooth11`

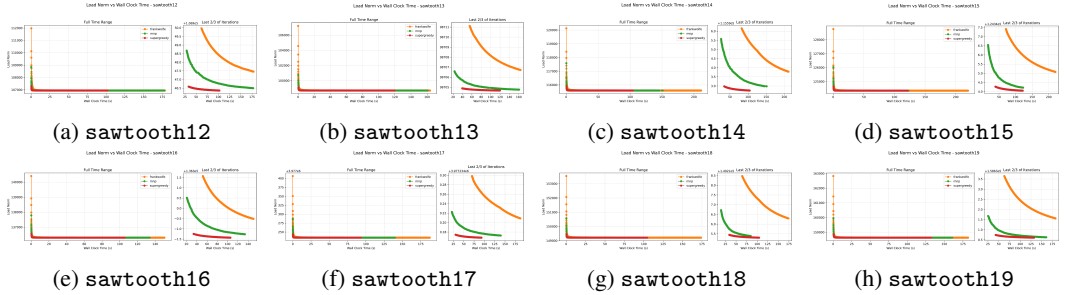

(a) `sawtooth12`    (b) `sawtooth13`    (c) `sawtooth14`    (d) `sawtooth15`

(e) `sawtooth16`    (f) `sawtooth17`    (g) `sawtooth18`    (h) `sawtooth19`

Figure 21: Min $s$-$t$ Cut load norm over time

## I.5    Generalized $p$-mean DSG

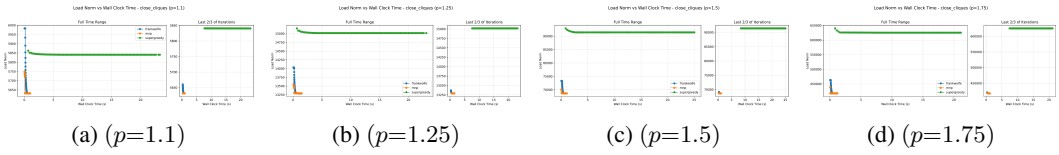

(a) ($p$=1.1)    (b) ($p$=1.25)    (c) ($p$=1.5)    (d) ($p$=1.75)

Figure 22: DSS load norm over time - `close_cliques`

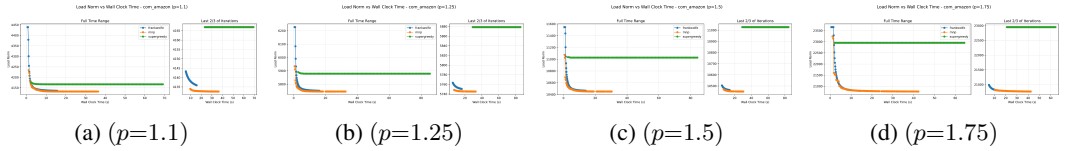

(a) ($p$=1.1)    (b) ($p$=1.25)    (c) ($p$=1.5)    (d) ($p$=1.75)

Figure 23: DSS load norm over time - `com_amazon`

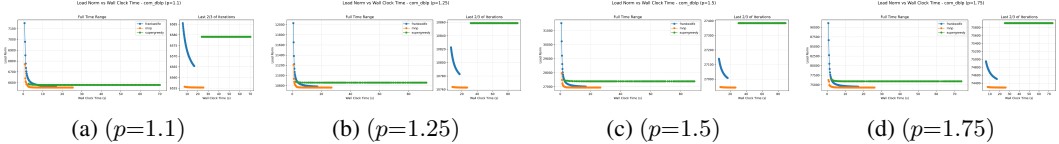

(a) ($p$=1.1)    (b) ($p$=1.25)    (c) ($p$=1.5)    (d) ($p$=1.75)

Figure 24: DSS load norm over time - `com_dblp`

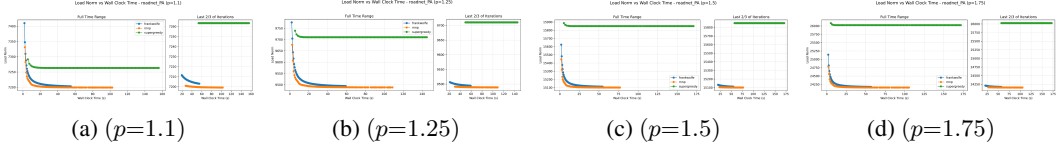

(a) ($p$=1.1)    (b) ($p$=1.25)    (c) ($p$=1.5)    (d) ($p$=1.75)

Figure 25: DSS load norm over time - `roadnet_PA`

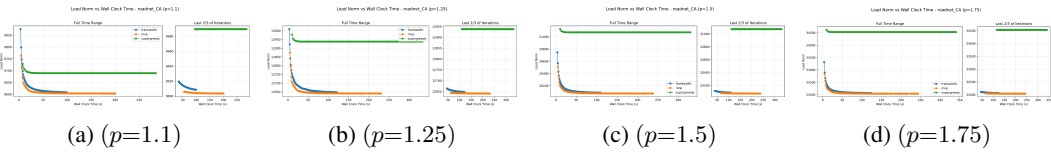

(a) ($p$=1.1)    (b) ($p$=1.25)    (c) ($p$=1.5)    (d) ($p$=1.75)

Figure 26: DSS load norm over time - `roadnet_CA`

## I.6 Contrapolymatroid Membership

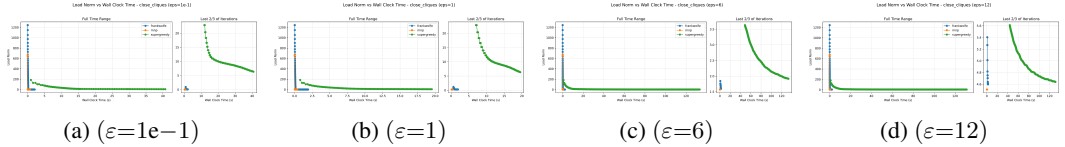

(a) $(\varepsilon=1\mathrm{e}{-}1)$      (b) $(\varepsilon=1)$      (c) $(\varepsilon=6)$      (d) $(\varepsilon=12)$

Figure 27: Contrapolymatroid Membership load norm over time - `close_cliques`

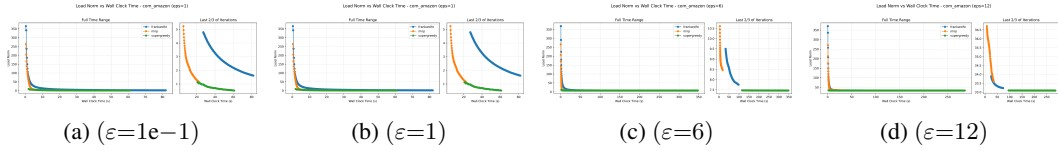

(a) $(\varepsilon=1\mathrm{e}{-}1)$      (b) $(\varepsilon=1)$      (c) $(\varepsilon=6)$      (d) $(\varepsilon=12)$

Figure 28: Contrapolymatroid Membership load norm over time - `com_amazon`

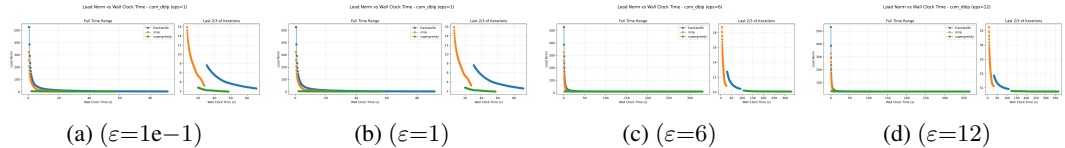

(a) $(\varepsilon=1\mathrm{e}{-}1)$      (b) $(\varepsilon=1)$      (c) $(\varepsilon=6)$      (d) $(\varepsilon=12)$

Figure 29: Contrapolymatroid Membership load norm over time - `com_dblp`

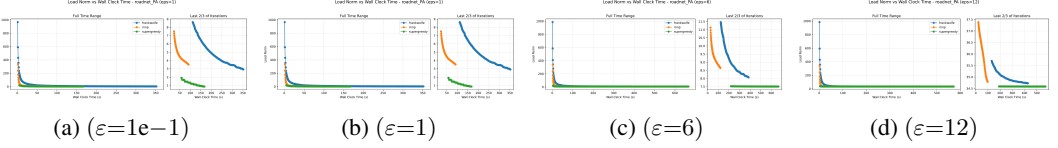

(a) $(\varepsilon=1\mathrm{e}{-}1)$      (b) $(\varepsilon=1)$      (c) $(\varepsilon=6)$      (d) $(\varepsilon=12)$

Figure 30: Contrapolymatroid Membership load norm over time - `roadnet_PA`

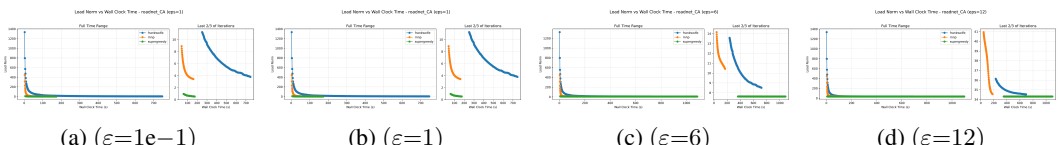

(a) $(\varepsilon=1\mathrm{e}{-}1)$      (b) $(\varepsilon=1)$      (c) $(\varepsilon=6)$      (d) $(\varepsilon=12)$

Figure 31: Contrapolymatroid Membership load norm over time - `roadnet_CA`

