# OpenReview forum: "Corporate Needs You to Find the Difference: Revisiting Submodular and Supermodular Ratio Optimization Problems"
_NeurIPS.cc/2025/Conference — NeurIPS 2025 spotlight_

### Official Review · Reviewer_z95E · 2025-06-06

**Clarity:** 3
**Significance:** 4
**Originality:** 4
**Rating:** 5
**Confidence:** 5

**Summary:**

In this paper, the authors study a broad range of combinatorial optimization problems related to submodularity and supermodularity, particularly focusing on submodular and supermodular ratio optimization problems. The authors introduce two novel optimization problems: Unrestricted Sparsest Submodular Set (USSS, Problem 5) and Unrestricted Densest Supermodular Set (UDSS, Problem 7). In USSS/UDSS, given a universe $V$ and a normalized submodular/supermodular function $f$, we are asked to find a nonempty $S\subseteq V$ that minimizes/maximizes $f(S)/|S|$. USSS is a generalization of the recently-introduced graph optimization problem called Heavy Nodes in a Small Neighborhood (HNSN, Problem 4), while UDSS is a generalization of Densest Supermodular Set (DSS, Problem 3), which is also known as a generalization of the well-known graph optimization problem called Densest Subgraph (DSG, Problem 2) and its variant called Anchored Densest Subgraph (ADS, Problem 6). In addition to the above ratio optimization problems, the authors consider two classic problems: Submodular Function Minimization (SFM, Problem 1) and Minimum Norm Point (MNP, Problem 8). The main contribution of this paper is to establish a novel algorithmic foundation for the above problems, especially focusing on USSS, UDSS, DSS, SFM, and MNP. To this end, the authors first prove that the five problems are equivalent in terms of exact optimization, meaning that any exact algorithm for one problem is applicable to any other problem. Specifically, they construct a strongly polynomial-time (at most $O(n\log n)$ time) reduction between any two problems. Note that some of the reductions are known or trivial (e.g., DSS to SFM) but many of them are not trivial at all. Furthermore, the authors also study the relationships among the problems in terms of approximation. Specifically, they demonstrate that MNP plays a key role in terms of additive approximation by proving that an additive approximation algorithm to MNP can be used to obtain additive approximate solutions to any other problems. Note that the additive approximation is unavoidable, since some of the problems induce negative objective values. Computational experiments using a vast amount of instances for the problems demonstrate that some algorithm for one problem that have never been considered as potential algorithms for the other problems outperform state-of-the-art methods specifically designed for those problems.

**Questions:**

- Is it possible to summarize the time complexities of the algorithms for different problems. For example, the reviewer is interested in the time complexity of SuperGreedy++ when applying it to SFM. Does it give a competitive complexity with state-of-the-art? The runtime is empirically thoroughly evaluated, but the reviewer believes this kind of work is useful.
- Isn't it worth studying restricted version of USSS, say SSS?

**Ethical Concerns:**

["NO or VERY MINOR ethics concerns only"]

**Final Justification:**

My initial assessment was to accept the manuscript. After carefully reviewing the authors’ responses and the discussions with the other reviewers, I find that the proposed revisions are reasonable. Therefore, I keep recommending acceptance.

**Limitations:**

yes.

**Quality:**

3

**Strengths And Weaknesses:**

**Strengths:**

This is a mathematically solid paper, with a lot of originality and significance. The authors succeed in establishing a novel algorithmic foundation for the diverse problems, including USSS, UDSS, DSS, SFM, and MNP, by constructing a strongly polynomial-time reduction between any two problems and approximation-preserving reductions from USSS, UDSS (DSS), and SFM to MNP. These types of reductions are quite beneficial, since each of the problems (or its special case) has very actively been studied in the literature and the reductions enable algorithmic cross-over, applying an algorithm developed for one problem to the others. For instance, the classic Fujishige--Wolfe's algorithm for MNP is now applicable to the other problems and the authors empirically demonstrate that when applying it to the recent HNSN, a special case of USSS, it significantly outperforms state-of-the-art methods. On the other hand, the recently-invented algorithm for DSS called SuperGreedy++ is now applicable to the classic SFM, and the authors again empirically demonstrate that SuperGreedy++ often performs better than state-of-the-art methods. The experiments are quite thorough, where a vast amount of instances for the problems are used and a lot of baseline methods are implemented and tested. Although the writing has some potential to be improved, the quality and clarity are generally satisfactory.

**Weaknesses:**

- The writing can be improved. Currently, the authors introduce USSS and UDSS as generalizations of the recently-introduced problems. However, this way makes the structure of the introduction a bit too complicated, where many special cases that are not directly studied in the methodological part are intertwined with the other main problems. The reviewer would suggest presenting the main problems first and then motivating them by putting some existing special cases. Considering the theoretical importance of the contribution, the reviewer believes the above presentation strategy still works while improves the readability a lot. The structure of the abstract can be modified in a similar way.

- It would be better to include proof sketches in the main body to make the readers get the intuition about the technical contents better.

- The selection of experiments presented in the main body should be justified. In the reviewer's understanding, the results for HNSN highlight the usefulness of the classic algorithms for the recent problem, while the results for Minimum $st$ Cut highlight the usefulness of the recently-invented algorithms for the classic problem.

- There are some missing references. ADS was originally introduced by Dai et al. (SIGMOD 2022) not by [31]. Nagano et al. (ICML 2011) studied an MNP-based algorithm for Densest $k$-Subgraph, which seems relevant to the present paper.

- There are several undefined notations, typos, and other minor things, listed below.

Missing references:
- Dai, Qiao, Chang: Anchored Densest Subgraph, SIGMOD 2022.
- Nagano, Kawahara, Aihara: Size-Constrained Submodular Minimization through Minimum Norm Base, ICML 2011.

Typos etc.:
- Line 9: MNP is undefined.
- Line 93: deg is undefined.
- Line 155: $n$ is undefined.
- Line 167: no data-structures -> no special data-structures
- Line 188: It would be better to highlight that the approximation-preserving reductions are different from the above reductions.
- Line 212: that each have -> each of which has
- Lines 222 & 247: Put a period.
- Line 226: $x(S)$ is undefined.
- Line 241: that that -> that
- Line 274: $n=|V|$ is redundant.
- Theorem 14: What is $d$? Should it be $x$? The operation $V-u$ is undefined.
- Line 310: Should $\delta(v)\subseteq S$ be $\delta(v)\cap S \neq \emptyset$?
- Figures 1b and 1d are not mentioned (in the main body).
- Line 384: Veldt -> Huang
- Line 401: Make a section for conclusion and limitations.

---

> ### Author Rebuttal · Authors · 2025-07-30
>
> # Response to Reviewer z95E
>
> Thank you for the detailed and thoughtful review. We are grateful for your recognition of the paper’s contributions. Below we respond to your helpful suggestions and questions:
>
> ---
>
> ### **Presentation and Writing**
>
> We appreciate your suggestions regarding the structure and clarity of the introduction and abstract. In particular:
>
> * We agree that beginning with the main problems (USSS, UDSS, DSS, SFM, MNP) and then introducing known special cases such as HNSN, ADS, and DSG as motivating applications could enhance readability; in fact, this was our original structure for the introduction. However, we found that starting with concrete problems first offered stronger motivation than beginning with the more abstract formulations. That said, we will revise the text to smooth the transition and introduce the broader abstract problems earlier to better guide the reader.
> * We also acknowledge that proof sketches in the main body would help with intuitions. Unfortunately, because of the page limit, we had to completely skip this for the submission. In the final version (when we get 1 additional page), we plan to include brief sketches for key reductions (e.g., DSS $\iff$ SFM, USSS $\Rightarrow$ MNP) and reference the full proofs in the appendix.
> * We will clarify the purpose of each experimental result more explicitly. Specifically, as you noted: (1) the HNSN results demonstrate the **relevance of classic MNP methods for new problems**, and (2) the s-t cut experiments show that **recent heuristics like SUPERGREEDY++ can compete with classical max-flow algorithms**. We’ll make this motivation more clear near Figures 1a–1d.
>
> ---
>
> ### **Missing References and Attribution**
>
> Thank you for catching this, we will correct the following:
>
> * **Anchored Densest Subgraph (ADS)**: We will cite the original work by **Dai et al., SIGMOD 2022** as the correct source for ADS.
> * **Nagano et al., ICML 2011**: This is indeed highly relevant. We will cite and discuss it in the revised related work section.
>
> ---
>
> ### **Notation, Typos, and Clarifications**
>
> We will address all the specific issues you noted, including undefined notations, missing periods, and inconsistencies in phrasing. These are extremely helpful. In particular:
>
> * Theorem 14: We will clean up the expression for the number of iterations (it should be $s-d$, not $s-x$) and fix the undefined operations.
>
> ---
>
> ### **Response to Questions**
>
> #### **Q1: Can you summarize the time complexities of the algorithms, especially SUPERGREEDY++ for SFM?**
>
>
> * For SFM, **SUPERGREEDY++** converges to a solution $S$ such that $f(S) \leq  \text{OPT} + 2n\varepsilon$ in
>
>   $$
>   \widetilde{O}\left(\frac{\alpha_f }{\epsilon^2}\right)
>   $$
>
>   iterations, where $\alpha_f$ depends on the function $f$ (From Theorem 14).
>
> Its runtime is still pseudo-polynomial due to the function-dependent terms. Empirically, it performs extremely well, but we agree the theoretical runtime should be compared with classical polynomial time algorithms like Iwata-Fleischer-Fujishige or Orlin’s algorithm.
>
> We will add a small table or section summarizing these complexities in the final version.
>
> #### **Q2: Is it worth studying restricted versions of USSS, like SSS?**
>
> Absolutely. This is a great idea and a natural extension. In fact, our original motivation for this paper; minimizing $f(S)/|S|$ for $f(S)=|N(S)|$ being the coverage function falls under SSS.
>
> While our paper focuses on **unrestricted** versions to allow non-monotone and negative functions (which are common in real-world applications like ADS and HNSN), **restricted versions** such as *Sparsest Submodular Set with Monotone or Positive f* could offer both:
>
> * Simpler theoretical analysis, and
> * Opportunities for improved approximations or faster algorithms.
>
> It is also worth noting that for DSS, one can obtain results in terms of multiplicative approximations that cannot be shown for UDSS. We have some preliminary results for SSS in similar vein - for example SuperGreedy performance in terms of curvature constant of $f$ or potentially convergence analysis for SuperGreedy++ that is similar
> to that for DSS in the Chekuri et al. SODA paper for a multiplicative $(1+\varepsilon)$-approximation rather than additive one.
>
> We will mention this direction explicitly in the conclusion as a promising avenue for future work.
>
> ---
>
> Once again, we thank the reviewer for this thoughtful, technically precise review. Your feedback directly improves the final version.

---

> > ### Comment · Reviewer_z95E · 2025-08-01
> >
> > Thank you very much for addressing my comments. I have carefully reviewed the authors’ response and found that all of the proposed revision plans are reasonable. I therefore maintain my original assessment and recommend acceptance.

---

### Official Review · Reviewer_Pm74 · 2025-06-30

**Clarity:** 3
**Significance:** 3
**Originality:** 4
**Rating:** 5
**Confidence:** 4

**Summary:**

This paper presents a unified theoretical and algorithmic framework connecting several submodular and supermodular ratio optimization problems, including Densest Subgraph (DSS), Heavy Nodes in a Small Neighborhood (HNSN),  Unrestricted Sparsest Submodular Set (USSS), and Submodular Function Minimization (SFM), Minimum Norm Point (MNP), etc. The authors establish strong polynomial-time equivalences among these problems and demonstrate that their approximation guarantees can be mutually preserved via reductions to the Minimum Norm Point (MNP) problem. This equivalence enables algorithmic cross-pollination: methods like SUPERGREEDY++, Frank-Wolfe, and Fujishige-Wolfe’s MNP algorithm—originally designed for specific settings—are shown to serve as universal solvers across all problem classes.

**Questions:**

1.	The paper states that DSG is polynomial time solvable. What about the rest problems mentioned in the paper? Like DSS, HNSN, USSS, etc.

2.	It seems that the condition of Theorem 13 is slightly different with the approximation result for MNP, no matter that we talk about additive approximation or multiplicative approximation. In my understand, the direct implication of the condition is that $||\hat{x}||_2\leq ||x^*||_2+\epsilon^2/||\hat{x}||_2$. So it might be useful to write down approximation solutions of MNP itself based on this condition in the theorem.

3.	I want to confirm whether the following understanding is correct. When we say the algorithm (SUPERGREEDY++, Frank-Wolfe, or Fujishige-Wolfe) can solve the problems mentioned in the paper efficiently and with good approximation, we mean that we should firstly transfer the problem into MNP , then use the algorithm to solve the MNP problem and finally transfer the solution back. It does not mean we can obtain good result if we apply the algorithm directly to the problem (suppose we have some natural way to apply the algorithm).

**Ethical Concerns:**

["NO or VERY MINOR ethics concerns only"]

**Final Justification:**

The results in the paper are interesting and important. The authors have adequately addressed my questions and clarified the issues mentioned in the review. However, due to page limitations, many details could not be fully elaborated. I look forward to seeing a more comprehensive journal version.

**Limitations:**

Yes

**Paper Formatting Concerns:**

One minor suggestion: submodular function minimization problem is trivial when the submodular function is monotone. However, just before the SFM definition in page 2, the paper introduces monotonicity. This is a little bit confusing.

**Quality:**

4

**Strengths And Weaknesses:**

Strengths:

1.	The connection between several submodular and supermodular ratio optimization problems is very interesting and important.

2.	The theoretical insights are compelling, particularly the formalization of how approximations for MNP transfer to other problems, which broadens the applicability of existing algorithms.

3.	The experimental results show the theoretical results can be applied in practice.

Weakness: The paper discusses many problems and algorithms, so due to the space limitation, it is difficult to cover every detail in the main body.

---

> ### Author Rebuttal · Authors · 2025-07-30
>
> # Response to Reviewer Pm74
>
> We thank the reviewer for their detailed and constructive feedback. Below we address the specific questions and suggestions:
>
> ---
>
> ### **Q1: Are the other problems (e.g., DSS, HNSN, USSS) also polynomial-time solvable like DSG?**
>
> Yes, all of the problems mentioned (DSS, USSS, UDSS, SFM, MNP) are, at least in theory, all **polynomial-time solvable in the exact optimization setting**. In fact, one of our main contributions (Theorem 12) is to show that these problems are **strongly polynomial-time equivalent**, with efficient reductions between them. This means that since Submodular Function Minimization (SFM) and MNP are known to be solvable in strongly polynomial time, so too are DSS, USSS, and others.
>
> Of course, as we also emphasize, the approximation landscape can vary wildly in practice, depending on the function structure, which is discussed in Theorem 13.
>
> ---
>
> ### **Q2: Theorem 13 condition vs. standard MNP approximation — can we derive an approximate solution to MNP itself from this condition?**
>
> The condition in Theorem 13 assumes an approximate solution $\hat{x} \in B(f)$ satisfying
>
> $$
> \|\hat{x}\|^2 \le \langle q, \hat{x} \rangle + \varepsilon^2 \quad \text{for all } q \in B(f),
> $$
>
> which upper bounds the **duality gap**. Theorem 14 guarantees such approximate solution after a number of iterations. While Theorem 13 focuses on how this approximate $\hat{x}$ leads to good *set-based* solutions for other problems (like SFM, USSS), it is true that this same $\hat{x}$ is also an **approximate solution to the MNP problem itself**, in the sense that $\hat{x}$ is close to the true minimum-norm point up to additive error $\varepsilon$.
>
> We appreciate the suggestion and will include a short clarification that this approximate solution serves both as a good MNP solution and as a bridge to solving the other problems.
>
> ---
>
> ### **Q3: Do we need to explicitly reduce to MNP before applying the algorithms? Or can we apply them directly?**
>
> The answer is: **both perspectives are valid**.
>
> * Our reductions show that all five problems can be *reduced to MNP*, and therefore, solving MNP (even approximately) suffices to solve the others.
> * However, we frequently apply the same algorithms (such as Frank-Wolfe, SUPERGREEDY++, or Fujishige-Wolfe) **directly** to each problem—meaning we obtain the solution to the target problem without explicitly transforming it into a Minimum Norm Point (MNP) instance. This is feasible because these algorithms operate over the same base polytope $B(f)$, which is common to all of these problem formulations.
>
> In fact, one of the key messages of the paper is that many of these algorithms are *implicitly solving MNP*, even when applied “naively” to problems like DSS or SFM. SUPERGREEDY++, for instance, was designed for DSS but is, under the hood, approximating the MNP solution in the base polytope.
>
> **Users can either reduce to MNP explicitly or apply these solvers directly if the base polytope access is available**, the behavior and guarantees remain the same.
>
> ---
>
> ### **Formatting Suggestion – Monotonicity and SFM**
>
> Thank you — this is a helpful comment. We will revise the paragraph to clarify that SFM applies **regardless of monotonicity**, and that the definition is simply introduced for completeness before defining monotone variants like DSS.
>
> ---
>
> Once again, we appreciate the thoughtful comments and high score.

---

### Official Review · Reviewer_XFrn · 2025-07-01

**Clarity:** 3
**Significance:** 4
**Originality:** 4
**Rating:** 6
**Confidence:** 2

**Summary:**

This work mainly proves five classes of problems to be equivalent, and also that approximate solutions to MNP can be transformed into approximate solutions for these five problems. This suggests that the method originally specifically on one of theseo another of these problems (namely, become somewhat problems can be transformed int universal), and even achieve a better performance. It also provides a possible way to connect the independent methods research together in a unified framework. To evaluate the discussion in the theoretical parts, the authors also conducted numerical experiments on a large-scale dataset, where basically convincing results are presented. Many interesting discoveries are discussed in the paper, while some of them are left as open questions in the limitations section.

**Questions:**

1. According to the experimental results, the authors claim that “MNP and FW excel on HNSN, they both underperform here, underscoring how problem structure affects solver efficiency.” (Line 371, “here” here refers to Minimum s-t Cut). If HNSN and MNP problems are already proven to be equivalent in Theorem 12 (Line 12), then why would the performances of two methos MNP and FW differ that much on two somehow equivalent problems? Can you provide any anaylsis of how exactly the problem structure is affecting the efficiency here?

2. According to the authors’ experimental results, in some problems the general methods can beat sota problem-specific methods. I'm quite curious about this result, does this mean the problem-specific method in fact does not take advantage of that much useful “specific information” in the problem? Is it possible to directly design problem-specific methods with better performance?

3. Some abbreviations are used repeatedly. e.g. MNP refers to both an algorithm (Fujishige-Wolfe Minimum Norm Point Algorithm, e.g. Line 372) and a problem class (MINIMUM NORM POINT Problem, e.g. Line 244). It’s not a major problem but can somehow affect the clarity of the paper, we expect that authors can do a little modification here.

**Ethical Concerns:**

["NO or VERY MINOR ethics concerns only"]

**Limitations:**

Yes. The authors have already discussed the main limitations from Line 401 to Line 405, though I’m also quite curious about the answers to these limitations.

**Paper Formatting Concerns:**

I do not see any issues.

**Quality:**

4

**Strengths And Weaknesses:**

Strength:
1. The text is clear and understandable, and the work is clearly discussed in the paper.
2. The idea of this paper is very interesting and of significance; it proves the equivalence of the problem types in Theorem 12, thereby creating an opportunity to unify the previously independent lines of research on these problems.
3. The experiments are well-conducted and are basically sufficient to numerically prove the ideas discussed in the theoretical part.

Weakness:
1. Some important limitations are left as open questions
2. Some abbreviations are used repeatedly

---

> ### Author Rebuttal · Authors · 2025-07-30
>
> # Response to Reviewer XFrn
>
>
> We sincerely thank the reviewer for their generous and thoughtful comments. We appreciate the recognition of the paper's contributions. Below, we respond to the specific questions and suggestions:
>
> ---
>
> ### **Q1: Why do MNP and FW perform well on HNSN but underperform on Minimum s-t Cut, despite theoretical equivalence?**
>
> This is an important and nuanced question. As you point out, **when solved exactly**, the problems are **mathematically equivalent**, and Theorem 12 provides the reductions showing how their optimal solutions can be transformed into one another. However, **in the approximate setting**, the situation becomes much more subtle. In approximation algorithms, problems with output values that are zero or non-negative are typically more challenging, as additive error—being the more appropriate measure in such cases—is harder to achieve, whereas multiplicative error is generally easier to handle for non-negative objectives.
>
>
> The key difference lies in the **structure of the underlying submodular functions**:
>
> * For the **HNSN** problem, the function $f(S) = |N(S)|$ is a **coverage function** — it is **monotone**, **non-negative**, and has a clean characterisations.
> * Conversely, in the **minimum s-t cut** formulation, the function $f(S) = |\delta(S \cup \{s\})| - |\delta(\{s\})|$ is **not monotone**, can take **negative values**, and typically leads to **more complex polytopes** with extreme points that are close together or highly skewed.
>
> These structural differences strongly influence the **convergence behavior of iterative solvers** such as Frank-Wolfe and MNP. In practice, we find that solver performance can vary significantly depending on the **geometry of the base polytope**, even when the underlying optimization problems are theoretically equivalent. For instance, in the **minimum s–t cut** formulation, the function $f(S) = |\delta(S \cup \{s\})| - |\delta(\{s\})|$ tends to be sensitive to small changes in the graph, which can result in a **"thin" or skewed base polytope** along certain directions.
>
> This geometric perspective is not merely anecdotal; some convergence analyses of Frank-Wolfe explicitly rely on notions like **pyramidal width**, a purely geometric quantity that captures how “sharp” or “flat” the polytope is in different directions. Since both Frank-Wolfe and MNP operate by navigating the **extreme points** of the base polytope, their efficiency is tightly coupled with its geometric structure.
>
> While we consistently observe that **at least one of our universal solvers performs well** on any given problem, **we do not yet fully understand** *which* solver performs best *when*, or *why*. Understanding the interplay between the function structure and solver behavior remains an open question, and is explicitly noted in the limitations of our work.
>
> ---
>
> ### **Q2: Why do general-purpose methods outperform problem-specific heuristics? Does this mean the heuristics aren’t leveraging the problem structure?**
>
> This is a great observation. In many problem-specific heuristics (e.g., greedy peeling or local ratio methods), the “specific structure” that is used is often **simplistic** or **myopic**, such as removing low-degree nodes without a global view.
>
> In contrast, general-purpose methods like MNP, Frank-Wolfe, and SUPERGREEDY++ implicitly operate on the **full combinatorial or polyhedral structure** of the problem. As a result, they can often **exploit hidden structure** better than naive heuristics.
>
> That said, it remains possible to design *better* problem-specific algorithms that leverage deeper properties (e.g., flow-augmenting paths, dual decompositions). Our results highlight that **some heuristics underperform not because the problem is hard, but because the heuristic doesn’t fully exploit the structure**, which we believe is an exciting direction for future algorithm design. We will elaborate on this in the final paper.
>
> ---
>
> ### **Q3: Ambiguity in the term "MNP" (problem vs. algorithm)**
>
> Thank you, this is a valid and very helpful suggestion. We agree that overloading "MNP" to mean both the **Minimum Norm Point problem** and the **Fujishige-Wolfe algorithm** can cause confusion. In the final version, we will explicitly distinguish:
>
> * *MNP (problem)*: the Minimum Norm Point problem
> * *FW-MNP*: the Fujishige-Wolfe algorithm used to solve MNP
>
> ---
>
> Again, we thank the reviewer for their encouraging assessment and insightful questions.

---

> > ### Comment · Area_Chair_CCci · 2025-08-07
> > **Please respond to author rebuttal**
> >
> > Dear Reviewer XFrn,
> >
> > Thanks a lot for reviewing this paper!
> >
> > It seems that you have not responded to the authors' rebuttal above. Given there are fewer than three days remaining for the author-reviewer discussion period, could you respond to it soon?
> >
> > Thanks!

---

> > ### Comment · Reviewer_XFrn · 2025-08-08
> >
> > Thank you for the reply to my previous concerns. I am quite happy with them. I, thereby, decided to keep my score as I gave a relatively high score in the first place.

---

### Official Review · Reviewer_xHD5 · 2025-07-03

**Clarity:** 4
**Significance:** 4
**Originality:** 4
**Rating:** 6
**Confidence:** 3

**Summary:**

Optimization problems involving submodular and supermodular
functions have many applications in computer science. The paper
addresses an important version of these problems, called the ratio
optimization problem where the goal is to optimize the ratio f(S)/|S|,
where S is the solution set and f is the submodular or supermodular
function. The paper points out that the ratio optimization problem models
a number of optimization problems studied in the literature. It is shown
that many of these problems are equivalent to each other (with respect to
optimal solutions) through polynomial time reductions. It is also shown
that traditional approximation methods (namely SuperGreedy++ and
Minimum Norm Point) yield good approximations for all of the problems
considered. Extensive experimental results are presented for the
problems considered.

**Questions:**

(1) In the definition of Problem 4 (lines 82--85), it seems to this
reviewer that the summation should be over the nodes in the solution
set S (rather than the set R of all nodes). This may just be a typo.
Please clarify.

(2) Lines 127--128: You mention that "... deemed impractical for
large graphs due to requiring an expensive binary search procedure".
Can you please explain? (Usually, binary search is not expensive.)

(3) As stated in Theorem 14, the number of iterations
depends on the maximum value of the submodular or supermodular function.
So, it seems to this reviewer that SuperGreedy++ is a pseudo-polynomial time
algorithm rather than a polynomial time algorithm. Can you please clarify?

A minor suggestion: In several places (e.g., lines 98, 105)
the paper has sentences of the form "[39] showed that ..." or
"indeed [31] state ...". It is better to revise them to something like
"Li et al. [39} showed that ...".

**Ethical Concerns:**

["NO or VERY MINOR ethics concerns only"]

**Final Justification:**

I have read all the reviews, rebuttals and the subsequent discussion. The results in the paper are nice and strong. I am glad that all the reviewers are in agreement regarding the quality of the paper. As mentioned by the Area Chair, I believe that this paper deserves to get a strong accept.

**Limitations:**

Yes.

**Quality:**

4

**Strengths And Weaknesses:**

Strengths:

(a) The topic of the paper (optimizing submodular and supermodular
functions) is of great interest to researchers because of numerous
applications.

(b) The paper nicely unifies known results for many problems and
shows that traditional approximation methods provide efficient
approximations for the problems. (The result for minimum st cut was
a pleasant surprise to this reviewer.)

(c) The experimental results are comprehensive.

(d) The paper is written very well.

Weakness:  This reviewer can't identify any major weakness. A (very)
minor weakness is that the the reader needs to keep track of a large
number of acronyms. A simple way to overcome this minor weakness
is to include a table of acronyms (in a longer version of the paper).

---

> ### Author Rebuttal · Authors · 2025-07-30
>
> # Response to Reviewer xHD5
>
> We sincerely thank the reviewer for their thoughtful and encouraging feedback.
>
> We address your comments and questions below:
>
> ### **(1) Clarification of Problem 4 (HNSN):**
>
> You are correct — this was a typo. The summation in the objective should indeed be over the solution set $S$, not over all of $R$. We will fix this typo in the revised version.
>
> ---
>
> ### **(2) Binary Search and Flow-Based Algorithms:**
>
> This is a valid question — binary search is efficient in general, and much faster than simply enumerating all possible density thresholds $\lambda$. Our intention was to indicate that earlier flow-based algorithms for densest subgraph problems used binary search over the density threshold $\lambda$, requiring a full **max-flow computation at each step of the search**. The need for **high-precision convergence and repeated max-flow computations** can makes this approach impractical at scale.
>
> For example, if we search for $\lambda$ in the range from $1/n$ to $n/2$ (the range of feasible densities), and aim for a precision of $\epsilon = 0.001$, the number of binary search steps required is roughly $\geq 30$ for $n\geq 2,000,000$. This translates to **30 full max-flow computations** for graphs with $n \sim 10^6$. Each such computation is costly in runtime and memory.
>
> In contrast, the **density-improvement method** (e.g., Huang et al. \[32], Hochbaum \[31]) iteratively updates $\lambda_k$ by solving
>
> $$
> \max_{S \subseteq V} \set{ f(S) - \lambda_k |S| },
> $$
>
> and typically converges in **fewer than 3-4 iterations**. This is because the number of "critical" values for $\lambda$ — points at which the optimal solution changes — is small in practice. Thus, density improvement "prunes" the solution space more quickly than sequential halving in binary search while achieving the same optimal result.
>
> We will clarify this example in the final version to better explain the practical difference.
>
> ---
>
> ### **(3) On Pseudo-Polynomial Time of SUPERGREEDY++ (Theorem 14):**
>
> This is an important observation. Indeed, the iteration bound in Theorem 14 depends on a function-dependent term involving the function $f$, and therefore **SUPERGREEDY++ is not guaranteed to run in strongly or weakly polynomial time** under our current analysis —  in that sense, you are absolutely correct that it is **pseudo-polynomial**.
>
> We would also like to point out that even in the original analysis of **SUPERGREEDY++** for the DSS problem by Chekuri et al. (SODA 2022), the convergence rate includes a term $ \Delta_f $ that is a pseudopolynomial in the input parameters (the function $f$). Our understanding of the precise convergence rate of SuperGreedy++ is still limited even when the function is non-negative and monotone. We will clarify this point in the revision.
>
> ---
>
> ### **(Minor Suggestion – Citation Style):**
>
> Thank you for the suggestion. We agree and will revise citation phrasings like “\[31] states” to “Huang et al. \[31] state” for better readability and consistency with the literature.
>
> ---
>
> ### **Additional Improvements:**
>
> Following your helpful suggestion, we plan to add a **table of acronyms** in the longer version to improve readability for new readers and reduce cognitive overhead.
>
> ---
>
> Once again, we truly appreciate your detailed and supportive review. We believe the revisions and clarifications above will further strengthen the clarity, accuracy, and impact of the paper.

---

> > ### Comment · Reviewer_xHD5 · 2025-08-05
> > **Thanks for your detailed response**
> >
> > I have read all the reviews and the responses from the authors. I thank the authors for going through my comments and providing detailed responses. The responses fully address all my concerns and I have no hesitation in recommending the acceptance of this paper.

---

### Decision · Program_Chairs · 2025-09-17

**Decision:**

Accept (spotlight)

**Comment:**

This paper presents a unified theoretical and algorithmic framework connecting several submodular and supermodular ratio optimization problems, such as Densest Subgraph (DSS), Heavy Nodes in a Small Neighborhood (HNSN), Unrestricted Sparsest Submodular Set (USSS), and Submodular Function Minimization (SFM), Minimum Norm Point (MNP). This paper establishes strong polynomial-time equivalences among these problems and demonstrates that their approximation guarantees can be mutually preserved via reductions to the Minimum Norm Point (MNP) problem. This equivalence enables algorithmic cross-pollination: methods like SUPERGREEDY++, Frank-Wolfe, and Fujishige-Wolfe’s MNP algorithm are shown to serve as universal solvers across all problem classes.

This paper is very well written, and its topic is interesting and important. It has nicely unified known results for different problems and shows that traditional approximation methods provide efficient approximations for the problems. The experiments are well-conducted and comprehensive.

All the reviewers recommend accepting this paper. The average rating for this paper is 5.5, and the minimum rating for this paper is 5. After reading the paper, the reviews, and the discussions, I agree with the reviewers that this paper is a significant contribution and strongly recommend accepting this paper.